# Interpretable LLM-based Table Question Answering

**Giang Nguyen**                                                    *nguyengiangbkhn@gmail.com*
*Auburn University*

**Ivan Brugere**                                                    *ivan.brugere@jpmchase.com*
*J.P. Morgan*

**Shubham Sharma**                                                 *shubham.x2.sharma@jpmchase.com*
*J.P. Morgan*

**Sanjay Kariyappa**[*]                                             *skariyappa@nvidia.com*
*NVIDIA*

**Anh Totti Nguyen**[†]                                            *anh.ng8@gmail.com*
*Auburn University*

**Freddy Lecue**[†]                                                *lecue.freddy@gmail.com*
*J.P. Morgan*

**Reviewed on OpenReview:** *https://openreview.net/forum?id=2eTsZBoU2W*

## Abstract

Interpretability in Table Question Answering (Table QA) is critical, especially in high-stakes domains like finance and healthcare. While recent Table QA approaches based on Large Language Models (LLMs) achieve high accuracy, they often produce *ambiguous* explanations of how answers are derived. We propose Plan-of-SQLs (`POS`), a new Table QA method that makes the model's decision-making process interpretable. `POS` decomposes a question into a sequence of atomic steps, each directly translated into an executable SQL command on the table, thereby ensuring that every intermediate result is transparent. Through extensive experiments, we show that: First, `POS` generates the highest-quality explanations among compared methods, which markedly improves the users' ability to simulate and verify the model's decisions. Second, when evaluated on standard Table QA benchmarks (TabFact, WikiTQ, and FeTaQA), `POS` achieves QA accuracy that is competitive to existing methods, while also offering greater efficiency—requiring significantly fewer LLM calls and table database queries (up to 25× fewer)—and more robust performance on large-sized tables. Finally, we observe high agreement (up to 90.59% in forward simulation) between LLMs and human users when making decisions based on the same explanations, suggesting that LLMs could serve as an effective proxy for humans in evaluating Table QA explanations. Code and data available at: `https://github.com/anguyen8/pos`

## 1 Introduction

An estimated 38% of office tasks involve working with tables, often using Excel (Richardson, 2022), highlighting the need for advanced tools for tabular data analysis. LLM-powered Table QA models (those in Fig. 1) address this gap by enabling users to quickly extract insights or answer questions for tables, making them invaluable in various industries. For example, financial analysts leverage these models to predict trends from tabular market data (Lo & Ross, 2024). Similarly, medical professionals use them to analyze tabular medical records of patients, facilitating accurate and timely treatment decisions (Bardhan et al., 2022).

---

[*]Work done at J.P. Morgan AI Research.
[†]Equal advising.

However, the value of these systems comes with high risks. Errors in financial decision making have led to catastrophic outcomes, such as the billion-dollar loss Citigroup faced in 2022 (Jane, 2024). In healthcare, the stakes are even fatal, considering a model that misjudges a man's health by overlooking his family history, resulting in his death from cardiac arrest weeks later (Stanford HAI, 2024). These examples underscore the pressing need for interpretability in Table QA to ensure safe and accountable use of AI (Fang et al., 2024).

Despite its importance, interpretability remains an underexplored dimension in Table QA literature. Recent approaches have significantly increased accuracy and often present themselves as *interpretable* solutions (Ye et al., 2023; Cheng et al., 2023; Wang et al., 2024), but this interpretability is unsubstantiated by empirical evidence. In practice, the explanations provided by these models can be unclear. For instance, as shown in Fig. 1**c**, a user cannot discern why certain rows were selected by a function **f_select_row()** or how an operation like **simple_query()** produced the final answer. In other words, current Table QA methods do not adequately explain their reasoning to users.

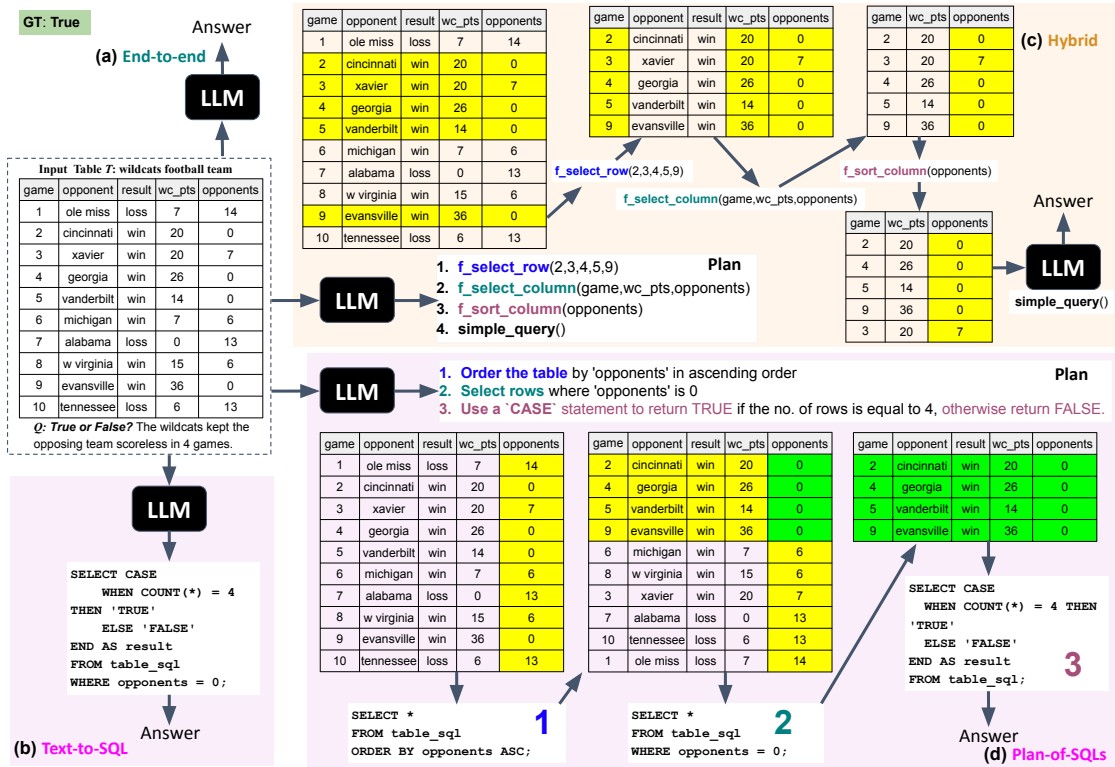

Figure 1: **(a)** End-to-End: relies entirely on an LLM to answer the question directly, leaving no room for users to understand the prediction. **(b)** Text-to-SQL: generates an SQL command to solve the question, requiring domain expertise to understand and becoming unintelligible when the question becomes complex. **(c)** Chain-of-Table or CoTable: performs planning with abstract functions and executes sequentially to arrive at the final answer. However, function arguments are not justified, and the final answer depends on the LLM's opaque reasoning. **(d)** Plan-of-SQLs or `POS` (Ours): plans in natural language, making each step simple and understandable. Each step is then converted into an SQL command, sequentially transforming the input table end-to-end to produce the final answer. We provide a public interface to compare explanations.

To address this gap, we propose Plan-of-SQLs (or `POS`), an LLM-based Table QA approach that places interpretability at its core. `POS` decomposes each question into a sequence of *atomic* steps, where each step is a simple sub-problem that can be translated into a corresponding SQL command and executed on the table. By design, each transformation is self-contained and limited in scope, forcing the model's decision-making to be broken down into transparent and verifiable steps. This has two key benefits. First, by requiring simple step-by-step table transformations via SQL, we avoid the model arbitrarily pulling in irrelevant data for answering. For example, existing methods that select a large subtable in one shot (Ye et al., 2023; Wang

et al., 2024) often include spurious entries due to the black-box LLM reasoning (see the irrelevant selection of `row3` in Fig. 1**c**). By contrast, **POS** uses programmatic conditions (e.g., "**WHERE** *opponents* = 0") to ensure only relevant entries are chosen (Step 2 of Fig. 1**d**). Second, **POS** largely avoids the opaque answer-generation step commonly found in LLM-based Table QA literature (Ye et al., 2023; Wang et al., 2024; Nahid & Rafiei, 2024b; Zhao et al., 2024) (illustrated in Fig. 1**a**,**c**). Rather than sending the entire table to an LLM, **POS** produces its final answer via a deterministic SQL query execution (Step 3 of Fig. 1**d**), making it entirely clear how the answer is derived from the data.

We thoroughly evaluate the interpretability and performance of **POS**. We compare against existing Table QA methods that provide explanations, including Text-to-SQL (Rajkumar et al., 2022), DATER (Ye et al., 2023), Chain-of-Table (CoTable)(Wang et al., 2024), and self-explaining (Madsen et al., 2024). Our interpretability evaluation spans three benchmarks—(1) explanation quality preference ranking (Ramaswamy et al., 2023; Yang et al., 2024), (2) forward simulation of the model's behavior (Doshi-Velez & Kim, 2017; Hase & Bansal, 2020; Chen et al., 2022; Mills et al., 2023), and (3) model prediction verification (Nguyen et al., 2021; Taesiri et al., 2022)—using both human participants and LLM-based judges. In all settings, **POS** usually yields the best results, showing that its explanations help users (human or AI) simulate and verify the model output far more effectively than the existing interpretable competitors. Furthermore, when tested on standard datasets (TabFact (Chen et al., 2020), WikiTQ (Pasupat & Liang, 2015), and FeTaQA (Nan et al., 2022)), **POS** achieves accuracy on par (within 1–2 points) with existing methods, while requiring drastically fewer LLM calls and database queries. Notably, **POS** scales robustly to large tables, where current methods Cheng et al. (2023); Ye et al. (2023); Wang et al. (2024) often struggle. Finally, we observe a high agreement between human evaluators and LLM-based evaluators (up to 90%) in forward simulation, suggesting that LLM-based proxies can reliably stand in for human judgment. In summary, our contributions are as follows.

- We introduce **POS**, a new Table QA method that is designed for interpretability. We carry out a thorough study of explanation effectiveness, and show that our explanations substantially improve users' understanding of the model's decision-making over existing interpretable methods (see Sec. 4.1.3).

- Compared to existing Table QA approaches, **POS** is more robust on large tables and significantly more efficient in its use of LLM calls and database queries (25× fewer LLM calls than DATER, and 25× fewer database queries than Binder), while maintaining competitive QA accuracy (see Sec. 4.2).

- Our experiments reveal high agreement between human users and LLM judges in evaluating Table QA explanations (up to 90.59% agreement in forward simulation). This suggests that LLMs could effectively proxy for human evaluators in evaluating Table QA explanations (see Sec. 4.1.5).

## 2 Related Work

### 2.1 Atomic Table Transformations for Table QA

LLM-based Table QA models have improved performance by decomposing complex input queries into smaller problems (Ye et al., 2023; Nahid & Rafiei, 2024b; Zhao et al., 2024) or by employing step-by-step reasoning (Wang et al., 2024; Wu & Feng, 2024; Abhyankar et al., 2024). However, these approaches often rely on *highly complex* table transformations—for instance, selecting a **question-relevant subtable** from a large input table via the opaque reasoning of LLMs (Ye et al., 2023; Nahid & Rafiei, 2024b; Wu & Feng, 2024; Abhyankar et al., 2024)—which can lead to errors in retrieving the correct data (see details in Appendix F). Often, the resulting sub-tables contain entries that are either irrelevant or logically incorrect with respect to the input question, due to the long-context challenges inherent in LLM reasoning.

In contrast, **POS** constrains a transformation of the table to an atomic SQL operation, that is, a single clause with at most one condition and one variable (e.g., "Select rows where `opponents` = 0"). This design has two main advantages. First, it improves accuracy and comprehensibility: because each step is minimal and focused, there is less room for LLMs to make mistakes, and the operation is easy for users to digest. Second, it enables fine-grained attribution: by executing each step with an atomic SQL query, we can pinpoint exactly which cells were used at that step, yielding a detailed trace of how the final answer is derived.

## 2.2 Program-based Table Transformations

Using programming languages such as SQL (Nahid & Rafiei, 2024b; Ye et al., 2023; Stoisser et al., 2025) or Python (Cheng et al., 2023; Chen et al., 2020) to manipulate tables is preferable for two reasons. First, these languages perform rule-based operations with explicit references to table cells, offering much greater traceability than the implicit, black-box transformations of LLMs (see contrastive examples between Fig. 1**a** & **d**). Second, programmatic transformations can handle large or complex table operations more reliably and efficiently, since they do not suffer from the context length and inconsistency issues that LLM-based methods encounter when processing entire tables (Chen, 2023; Wang et al., 2024; Nahid & Rafiei, 2024b).

**POS** builds on this line of work by leveraging program-based transformations—specifically, SQL—to solve Table QA. Notably, **POS** uses SQL *exclusively* for executing reasoning steps. To our knowledge, only two methods in the Table QA literature, LPA (Chen et al., 2020) (using Python-Pandas) and Text-to-SQL (Rajkumar et al., 2022), handle queries *end-to-end* through program-based operations. However, since Text-to-SQL generates a single complex SQL command for an input question (Fig. 1**b**), it requires a highly powerful Text-to-SQL model and often generates error-prone commands (Shi et al., 2020). Similarly, LPA's one-pass program synthesis can be brittle, as generating a correct multi-step program in one go is inherently challenging (Chen et al., 2020). In contrast, **POS** breaks the problem into multiple simpler SQL queries corresponding to atomic sub-steps initially expressed in natural language. This stepwise use of SQL removes the need for an advanced program generator and, as our results will show in Sec. 4, leads to higher overall accuracy and interpretability than Text-to-SQL or LPA.

## 2.3 Evaluating Interpretability

Interpretability is a critical aspect of AI models, and there is a rich body of work on evaluating explanations with human users (Adebayo et al., 2020; Nguyen et al., 2021; Kim et al., 2022; Taesiri et al., 2022; Colin et al., 2022; Steyvers & Kumar, 2023; Chen et al., 2023a; Nguyen et al., 2024a;b; Zhang et al., 2025). In the context of LLM-based Table QA, TAPERA (Zhao et al., 2024) had users subjectively rate the *faithfulness* and *comprehensiveness* of explanations on a Likert scale. However, such ratings serve only as proxies for explanation quality and do not directly measure the explanations' utility to users. **POS** guarantees 100% faithfulness (the explanation steps exactly produce the answer) and comprehensiveness (no reasoning step is hidden) because its explanations explicitly represent the model's reasoning through actual executed SQL operations. Therefore, in our experiments, we prioritize direct evaluation of explanations' effectiveness—such as how well users can simulate or verify the model's behaviors—over faithfulness or completeness metrics.

# 3   POS: Interpretable Table QA with Atomic Table Transformations

**Problem Formulation.**   In Table Question Answering (Table QA), each sample is represented as a triplet $(T, Q, A)$, where $T$ is a table, $Q$ is a question about the table, and $A$ is the answer. The task is to predict the answer $A$ given the question $Q$ and the table $T$. **POS** decomposes $Q$ into steps $\rightarrow$ converts each step into an SQL command $\rightarrow$ applies these commands sequentially to $T$ to arrive at $A$.

**Atomicity in Table QA Reasoning.**   We define an *atomic step* as a simple, minimal natural-language table operation that can be reliably translated into an SQL query. Specifically, each operation is restricted to: (i) at most one condition (e.g., $=$), and (ii) at most one column or variable in that condition (e.g., `opponents`). Enforcing this atomicity keeps each step's translation and execution reliable and lowers the chance of errors. It also makes the model's reasoning more interpretable: because each step addresses only a small part of the problem, a human can easily understand the purpose and effect of that step.

## 3.1 Planning in Natural Language

Rather than planning in a space of abstract functions for Table QA like Chen et al. (2020) and Wang et al. (2024), we perform the reasoning-step generation in *natural language*. Leveraging an LLM to plan in natural language takes advantage of the model's strong priors from its language training (Huang et al., 2022). It also makes the plan inherently understandable: each step is described in natural language, which is far more interpretable than, say, a sequence of function names (e.g., simple_query()) whose purpose is not

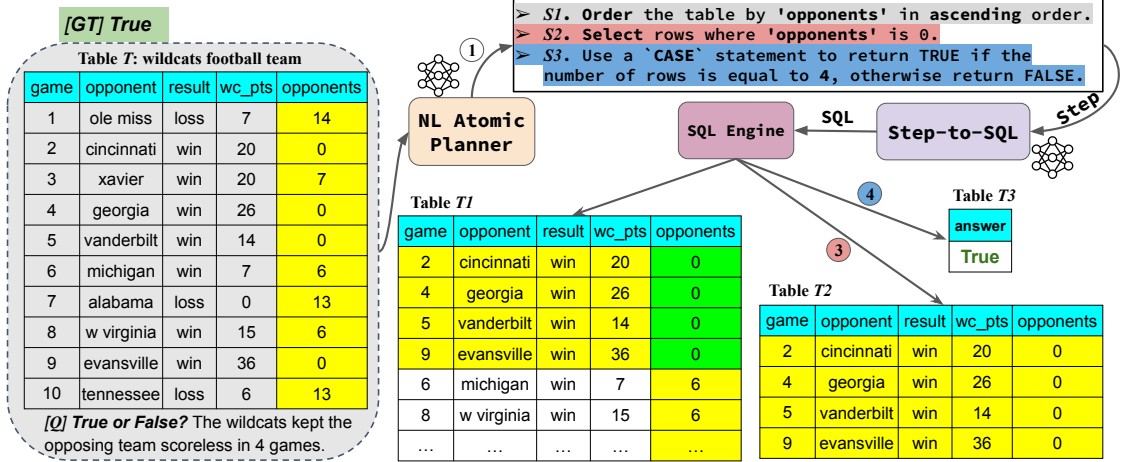

Figure 2: Illustration of Plan-of-SQLs (`POS`). ① The **Natural Language (NL) Atomic Planner** takes $(T, Q)$ as input and generates a step-by-step plan in plain language. ② **Step-to-SQL** then takes $(T, S_1)$ and converts the first step $S_1$ into an SQL query, which is executed on $T$ to produce an intermediate table $T_1$. ③ **Step-to-SQL** takes $(T_1, S_2)$ to produce and execute the next SQL query, yielding $T_2$. ④ The final **Step-to-SQL** uses $(T_2, S_3)$ to generate an SQL query that returns the final answer. We provide an interactive demo interface of `POS`.

immediately clear (Wang et al., 2024). In Fig. 2–①, the **NL Atomic Planner** takes the input $(T, Q)$ and produces a plan of atomic steps in natural language that outlines how to derive the answer $A$. `POS` prompts an LLM with examples and explicit instructions to incorporate the atomic decomposition. Specifically, we use a prompt of this form:

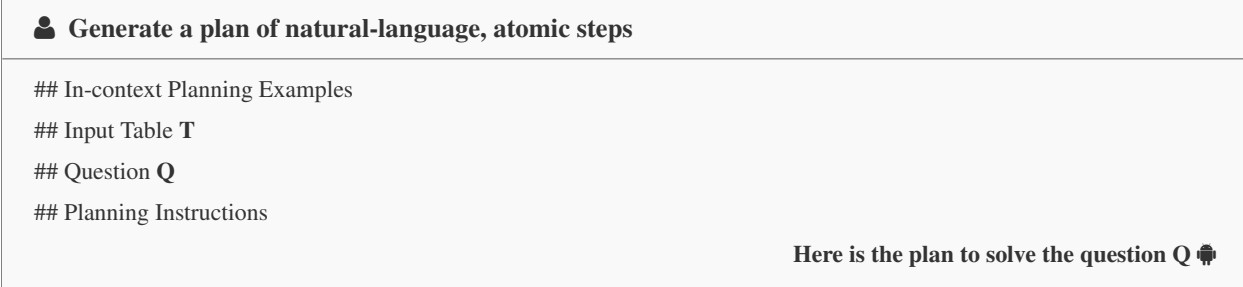

### 3.2 Converting Step to SQL

Once we have the plan in natural language, the next stage is to translate each step into an executable SQL query. We leverage the LLMs' capability as a Text-to-SQL converter (Hong et al., 2024) to perform this conversion for each step. There are two main reasons to use a dedicated **Step-to-SQL** module (Fig. 2). First, executing the reasoning steps via SQL commands greatly improves the correctness of each step compared to letting LLMs manipulate the table implicitly in its hidden state (see our analysis in Appendix C). In particular, offloading the computation to an SQL query ensures each operation is carried out exactly and eliminates errors from the LLMs' reasoning. Second, using SQL makes the sequence of transformations fully trackable: for every step, we know precisely which table entries were accessed or modified, since the SQL query specifies those conditions. This allows us to identify what information each step used.

In Fig. 2, the **Step-to-SQL** module takes the current intermediate table (initially the original table $T$) and one atomic step as input, and outputs the corresponding SQL query. We steer an LLM to generate an SQL with a specialized prompt template, as illustrated below:

---

👤 **Convert an atomic step into SQL**

## In-context Step-to-SQL Examples

## Current Intermediate Table

## Natural-Language Atomic Step

## Step-to-SQL Instructions

**Here is the SQL to execute the step** 🤖

---

After converting all steps, **POS** executes them sequentially using an SQL engine (we use SQLite3 (Gaffney et al., 2022) in our implementation). The intermediate result of each SQL command becomes the input for the next step, faithfully carrying out the transformations specified by the plan (see the chain Fig. 2– ②→③→④). In contrast to end-to-end or "chain-of-thought" Table QA methods that rely on an LLM's latent reasoning to jump to the final answer (as in Fig. 1**a**, **c**), **POS** maintains interpretability throughout the process: the answer is obtained through another SQL operation whose effects can be understood.

### 3.3 Generating Explanations for Table QA

Despite the need for interpretability, prior work has not established a formal method to generate human-understandable explanations for LLM-based Table QA models. Most studies focus on improving accuracy and leave the model's decision-making process opaque (Wu & Feng, 2024; Kong et al., 2024; Cheng et al., 2025). To bridge this gap, we propose an approach to generate explanations for Table QA using attribution maps—the main medium for explaining AI decisions to humans in various domains, including image classification (Colin et al., 2022), text analysis (Hase & Bansal, 2020), and time series (Theissler et al., 2022).

For each step, we create an attribution map on the intermediate table that highlights the information used in that step. Specifically, as we apply a transformation, we mark the rows and columns that were selected or affected in  yellow , and we highlight the cells that satisfy the step's condition in  green . By doing this for every step (see Fig. 2), we obtain intermediate tables annotated with highlights indicating which data contributed to the final answer. Finally, we present the explanation as a *chain of attribution maps*: each step is shown alongside its highlighted intermediate table. This allows a user to visually follow the process of reasoning. In practice, the explanation shows *which* cells were used *when*, making it clear *why*, for example, the final answer is True in the running example of Fig. 2. We also apply this visualization approach to other Table QA methods, such as CoTable (Wang et al., 2024) and DATER (Ye et al., 2023), with additional examples provided in Appendix A.

## 4 Experiments

We conduct experiments using three popular and standard Table QA benchmarks: TabFact (Chen et al., 2020), WikiTQ (Pasupat & Liang, 2015), and FeTaQA Nan et al. (2022). **TabFact** is a fact verification dataset in which each statement associated with a table is labeled TRUE or FALSE. We use the cleaned TabFact dataset from Wang et al. (2024) and evaluate Table QA methods with binary classification accuracy on the 2,024-sample test-small set. **WikiTQ** is a question-answering dataset where the goal is to answer human-written questions using an input table. Using the dataset and evaluation scripts from Ye et al. (2023), we assess model denotation accuracy (whether the predicted answer is equal to the ground-truth answer) on the 4,344-sample standard test set. **FeTaQA** is a free-form Table QA dataset where the task is to generate free-form natural language responses based on information retrieved or inferred from a table. We evaluate models on the 2,003-sample standard test set using BLEU and ROUGE.

### 4.1 Evaluating Explanations in Table QA

**Baselines.** We select Text-to-SQL (Rajkumar et al., 2022), DATER (Ye et al., 2023), and Chain-of-Table (CoTable) (Wang et al., 2024) to benchmark the interpretability of **POS**. They are chosen for their high performance, interpretability, and reproducibility (see details of baseline methods in Appendix A). Later, we

compare `POS` with self-explanations (Madsen et al., 2024), where LLMs are prompted to provide explanations for their own answers.

**Evaluation measures.** To evaluate the quality of explanations, we involve both human judgments and LLM-based judgments (which we denote as LLM-as-XAI-judge). We follow the two complementary evaluation perspectives (qualitative and quantitative) proposed by Doshi-Velez & Kim (2017).

In the *qualitative* evaluation, the judge is shown the table, the question, the model's answer, and the model's explanation. The task is to rank the explanations from different methods by overall perceived preference (to identify which explanations users are most likely to use in practice). This preference rating is based on a clearly defined rubric comprising three criteria: clarity, coherence, and usefulness in understanding the model's reasoning. We refer to this as a **Preference Ranking** task, following prior works on comparative explanation evaluation (Ramaswamy et al., 2023; Yang et al., 2024; Zhang et al., 2025; Wazzan et al., 2025). For each table question, we collect a ranking of the four methods' explanations (1 = best, 4 = worst). To ensure fairness, we only consider test samples where all methods being compared got the question either correct or incorrect (so that judges are comparing explanations for answers of the same correctness). We aggregate rankings over 707 TabFact questions meeting this criterion, computing the average rank for each method (lower is better ↓).

In the *quantitative* evaluation, the judge is asked to perform tasks that objectively measure how well the explanation informs them about the model's behaviors. We use two standard tasks for evaluating explanations: *Forward Simulation* (Doshi-Velez & Kim, 2017; Hase & Bansal, 2020; Chen et al., 2022; Mills et al., 2023) and *Model Prediction Verification* (Nguyen et al., 2021; Taesiri et al., 2022; Chen et al., 2023a).

In **Forward Simulation**, the judge is given the table, question, and explanation *without the model's answer*, and must predict what answer the model's output would be. This evaluates how clearly the explanation communicates the model's decision boundary or reasoning process to users.

In **Prediction Verification**, the judge is given the table, question, model's answer, and explanation, then must decide whether the model's answer is correct or not based on the explanation. This measures how well the explanation justifies the model's prediction (e.g., can the judge catch model errors using explanations?).

We compute performance for these tasks as the percentage of samples in which the judge makes a correct decision, reported as Simulation Accuracy and Verification Accuracy, respectively.

### 4.1.1 Evaluating explanations with human users

**Motivation.** Human evaluation is considered the gold standard for evaluating AI explanations, as humans are the ultimate users who work with AI models (Doshi-Velez & Kim, 2017). We aim to study how explanations help humans in understanding then predicting model behaviors via Forward Simulation.

**Participants and Data.** We recruit 32 volunteers, all of whom are undergraduate, master, or Ph.D. students in Computer Science. In each session, users select one of four explanation methods and complete 10 samples, with an option to participate in multiple sessions. We collect 800 responses ($\approx$ 200 per method).

### 4.1.2 Evaluating explanations with LLMs

**Motivation.** The use of LLMs trained to align with human preference (Ouyang et al., 2022) as judges has been gaining attention due to their strong correlation with human judgments (Dubois et al., 2024; Zheng et al., 2023; Liu et al., 2023; Mills et al., 2023; Fernández-Becerra et al., 2024; Poché et al., 2025). This makes LLM judge a promising, scalable solution for evaluating explanations, particularly in tasks like Table QA, where the information is still text-based yet structured. Thus, we are motivated to leverage LLM judges for all three tasks: Preference Ranking, Forward Simulation, and Model Prediction Verification.

**LLM judge.** Inspired by recent works showing the effectiveness of OpenAI's instruction-tuned GPT models as reliable judges (Zheng et al., 2023; Liu et al., 2023; Dubois et al., 2024), we utilize 3 OpenAI's LLMs: `gpt-4-turbo-2024-04-09`, `gpt-4o`, and `gpt-4o-mini` to evaluate Table QA explanations. We ensure the prompts encourage the model to follow the given instructions strictly (see Appendix G for the exact prompt templates and calibration procedures). We also explore the use of open-source LLMs as judges in Sec. 4.1.4.

Table 1: (a) Preference Rankings for explanation methods given by LLM judges on TabFact. Lower values indicate better rankings ↓. (b) Simulation Accuracy ↑ (%) of LLM and human judges on TabFact.

| XAI method | (a) Preference Ranking (1 → 4) ↓ | | | | (b) Simulation Accuracy (%) ↑ | | | |
|---|---|---|---|---|---|---|---|---|
| | Text-to-SQL | DATER | CoTable | POS | Text-to-SQL | DATER | CoTable | POS |
| GPT-4 | 3.33 | 3.36 | 1.98 | **1.33** | 75.15 | 80.04 | 79.99 | **84.89** |
| GPT-4o-mini | 3.95 | 2.75 | 1.75 | **1.55** | 65.67 | 73.57 | 76.53 | **81.61** |
| GPT-4o | 3.60 | 3.35 | 2.04 | **1.01** | 73.73 | 78.21 | 79.55 | **85.25** |
| Human | - | - | - | - | 83.68 | 86.50 | 84.29 | **93.00** |

Table 2: Verification Accuracy ↑ (%) of LLM judges on TabFact and WikiTQ.

| XAI method | TabFact | | | | WikiTQ | |
|---|---|---|---|---|---|---|
| | Text-to-SQL | DATER | CoTable | POS | DATER | POS |
| GPT-4 | 49.93 | 57.56 | 60.38 | **72.08** | **73.50** | 72.38 |
| GPT-4o-mini | 55.37 | 55.43 | 61.36 | **76.74** | 64.58 | **71.93** |
| GPT-4o | 55.97 | 70.95 | 67.34 | **72.85** | 73.31 | **74.45** |

### 4.1.3 Findings from the evaluation of explanations

**POS is ranked highest in quality.** In Tab. 1(a), POS explanations consistently receive the best ranks from all LLM judges. Specifically, our explanations achieve average ranks of 1.33, 1.55, and 1.01 from GPT-4, GPT-4o-mini, and GPT-4o; respectively, substantially outperforming CoTable, DATER, and Text-to-SQL.

This shows that POS explanations are regarded by LLM judges as the clearest, most coherent, and most helpful for understanding the Table QA model's reasoning process (see the rubrics in Appendix G). In practice, this preference could translate into increased trust—a key factor for AI adoption in high-stakes domains (Doshi-Velez & Kim, 2017).

**POS is most effective for predicting model behaviors.** Tab. 1(b) shows that POS effectively helps human and LLM judges predict the model behaviors. Specifically, human judges achieve 93.00% with POS explanations, outperforming other methods such as DATER (86.50%) and CoTable (84.29%). Similarly, across all LLM judges, POS consistently yields the highest accuracy, with improvements ranging from 5% → 6% over the second best methods.

**POS is most effective for model prediction verification.** We perform this experiment on TabFact and WikiTQ, comparing the verification accuracy with different explanation methods in Tab. 2. We find that POS is the best method in five out of six settings. In addition, the improvements between DATER and POS are more pronounced in TabFact compared to WikiTQ, suggesting that the effectiveness of explanations is influenced by the nature of the task. As a more complex dataset, WikiTQ makes it inherently more difficult for the judges to verify the predictions. Please note that we do not compare POS with Text-to-SQL and CoTable on WikiTQ, as these methods have not been publicly released for this benchmark.

Later in Sec. 4.1.6, we present an ablation study that further studies the factors driving POS's improved interpretability.

**Qualitative rankings strongly correlate with quantitative measures.** Using Tab. 1 & Tab. 2, we perform a correlation analysis to study whether qualitative preference rankings inform quantitative measures. Since lower rankings in Tab. 1(a) indicate higher-quality explanations, we invert the rankings to align higher simulation/verification accuracy with higher preference.

Interestingly, we find *statistically significant positive* Pearson correlations between preference rankings vs. simulation accuracy ($r = 0.7865$, $p = 0.0024$) and vs. verification accuracy ($r = 0.7035$, $p = 0.0107$). These high correlations suggest that in Table QA, our proposed rubrics based on perceived quality (see Appendix G) can effectively identify high-utility explanations. This allows for efficient pre-selection of explanations to present to users, significantly reducing the need for expensive and time-consuming user studies.

**Comparison with post-hoc self-explanations.** Finally, we investigate whether a simple, post-hoc self-explaining approach could rival POS in interpretability. Tab. 10 compares POS with a widely used "self-explanation" method, in which the LLM first generates an answer and then retrospectively explains its answer (see an example in Fig. 8).

We find that self-explanation remains substantially less effective than POS in all three interpretability benchmarks, largely due to the *lack of faithfulness* in post-hoc explanations (Madsen et al., 2024; Chen et al., 2023b; Agarwal et al., 2024). By contrast, POS grounds its explanations in offline transformation steps (via SQL execution), yielding absolute faithfulness and comprehensiveness, which are especially valuable for model simulation or verification. We provide more details in Appendix A.3.

### 4.1.4 Evaluating explanations with open-source LLMs

So far, our evaluation relies on GPT-family models. This can restrict the generalizability of our findings in Sec. 4.1.3. To address this, we use two additional model families—**Qwen** and **Llama**—to confirm whether the observed trends in explanation quality persist across multiple LLM families.

We use TabFact as the benchmark dataset, focusing on two tasks: Forward Simulation and Model Prediction Verification. In both tasks, we test four methods—Text-to-SQL, DATER, CoTable, and POS —under open-source **Qwen2.5-72B-Inst** and **Llama-3.1-405B-Inst** hosted by SambaNova[1].

Table 3: Evaluating Table QA explanations (%) on TabFact with open-source models.

| XAI Judge | Text-to-SQL | DATER | CoTable | POS |
|---|---|---|---|---|
| *Forward Simulation Accuracy* | | | | |
| Qwen2.5-72B | 84.31% | **91.38**% | 84.18% | 90.59% |
| Llama-3.1-405B | 79.08% | 86.63% | 79.10% | **90.59**% |
| Human | 83.68% | 86.50% | 84.29% | **93.00**% |
| *Model Prediction Verification Accuracy* | | | | |
| Qwen2.5-72B | 71.57% | 75.49% | 79.25% | **80.01**% |
| Llama-3.1-405B | 75.30% | 77.57% | 77.08% | **78.49**% |

We report the evaluation results on open-source LLMs in Tab. 3. In Verification, POS consistently delivers top performance, while Text-to-SQL remains the worst across all model families. In Forward Simulation, POS outperforms other baselines on Llama, although DATER slightly exceeds POS with Qwen2.5 (91.38% vs. 90.59%). In general, we find that the interpretability provided by POS translates well to open-source LLMs.

### 4.1.5 LLM–Human agreement in XAI evaluation

In Tab. 4, we report the instance-level *agreement* between LLM-based and human forward simulations *on the same samples* for four explanation methods. This metric differs from accuracy; it measures how often LLM judges and humans arrive at the same decision when given identical information (i.e., the input and an explanation). We find that: First, POS consistently yields the highest LLM–human agreement across all tested LLM judges, reaching up to 90.59%. This suggests that when explanations are faithful and grounded—e.g., via atomic SQL steps—both LLMs and humans converge on similar decisions in evaluating explanations. Second, baseline methods exhibit lower agreement (71–83%), likely because their explanations are less informative or less faithful, leading LLM judges and humans to follow different reasoning paths. Practically, these high agreements (71–90%) suggest that LLMs may serve as effective proxies for early-phase evaluation of Table QA explanations, reducing the high cost of user studies while still approximating human judgments.

### 4.1.6 Ablation study on POS interpretability

To better understand which components of POS (Sec. 3) most strongly contribute to its interpretability, we perform an ablation study on both TabFact and WikiTQ. We focus on the model prediction verification in which an XAI judge (here, GPT-4o-mini) is shown POS's explanations and asked to verify the final answer.

---

[1]https://sambanova.ai/

Table 4: LLM–human agreement (%) on forward simulation for TabFact. Each cell indicates how often an LLM's decisions align with the human decisions on the same subset of samples (listed in Human Samples). Higher values suggest stronger agreement between LLMs and humans.

| Method | GPT-4o-mini | Qwen2.5-72B-Inst | Llama-3.1-405B-Inst | Human Samples |
|---|---|---|---|---|
| Text-to-SQL | 71.24% | 79.08% | 75.16% | 153 |
| DATER | 81.98% | 83.14% | 76.74% | 172 |
| CoTable | 79.10% | 82.49% | 80.79% | 177 |
| **POS** | **88.30**% | **90.59**% | **89.41**% | 171 |

We remove (i.e., ablate out) one of the following three core components in POS and observe the resultant drop (or change) in verification accuracy: (**i**) Atomic operations: Instead of enforcing single-condition transformations, we allow complex multi-condition steps. (**ii**) NL planning: We replace the natural-language planning with a direct prompt that asks the LLM to generate a sequence of SQL commands to solve the question. (**iii**) SQL execution: We replace the SQL-based transformations with direct black-box LLM-based transformations.

Tab. 5 shows the verification accuracy in each ablation setting. We find that removing SQL execution leads to the biggest interpretability drop (from 76.74 to 64.19% on TabFact), while removing the natural-language planning also leads to a substantial decrease. Atomic operations, though beneficial, exhibit the smallest impact when removed.

Table 5: Verification Accuracy (%) on TabFact and WikiTQ. Removing each component from POS reveals its impact on interpretability: SQL execution is most critical, followed by the plan, and then atomic operations.

| Variant | TabFact | WikiTQ |
|---|---|---|
| POS (Full) | 76.74% | 71.93% |
| — Atomic operations | 76.65% (-0.09) | 71.74% (-0.19) |
| — NL planning | 67.96% (-8.78) | 66.20% (-5.73) |
| — SQL execution | 64.19% (-12.55) | 66.54% (-5.39) |

By translating each atomic step into an explicit SQL query, POS naturally produces step-by-step *attribution maps* that pinpoint exactly which table cells influence the final answer. This clarity is especially valuable for verification: if a highlighted entry in the explanation is irrelevant or obviously mismatched to the question, a user can readily infer that the model's final answer is suspect. The natural-language plan itself also matters significantly, presumably because it makes the chain-of-reasoning much easier to follow. Although ablating atomicity causes only a minor drop—likely because modern LLMs already employ strong step-by-step reasoning priors (Wei et al., 2022)—we argue that explicit atomic steps remain meaningful for human understanding. Please refer to Appendix D for qualitative examples.

## 4.2 Evaluating Table QA Performance

**Baselines.** We compare POS with several baseline methods, categorizing them into three groups based on *how table transformation and answer generation are performed*: LLM-only, program-only, and hybrid approaches. Unless otherwise noted, we use a `temp = 0` and `top-p = 1` for LLM generation.

### 4.2.1 Table QA for TabFact and WikiTQ

As shown in Tab. 6, POS achieves strong performance on both TabFact and WikiTQ on two different LLM backbones. When paired with GPT-3.5, POS achieves 78.31% accuracy on TabFact and 54.80% on WikiTQ, yielding substantial gains over LLM-only methods such as End-to-End QA, Few-Shot QA, and Chain-of-Thought. It also surpasses other program-only baselines by wide margins: for instance, POS outperforms Text-to-SQL by +13.6 points and LPA by +9.41 points on TabFact. Compared to hybrid approaches, which combine LLM and program-based operations, POS offers a compelling alternative. Although POS scores lower accuracy on both benchmarks than these hybrid methods, its exclusive use of program-based transformations ensures complete transparency of each reasoning step, making it much easier for users to verify and understand the underlying decision-making process.

Table 6: Accuracy (%) for TabFact and WikiTQ using GPT-3.5 and GPT-4o-mini. "Decomposed" indicates whether queries are decomposed into sub-problems (Fig. 2–①). "Transformed by" refers to whether intermediate tables are transformed by an LLM or a program (Fig. 2–②–③). "Answered by" specifies whether the final answer is generated by an LLM or a program (Fig. 2–④). LLM-only approaches provide the final answer without table transformations. The best performance for each model and dataset is shown in **bold**.

| Method | Accuracy (%) | | Decomposed | Tables transformed by | Final answer by |
| | TabFact | WikiTQ | | | |
|---|---|---|---|---|---|
| GPT-3.5 (`gpt-3.5-turbo-16k-0613`) | | | | | |
| End-to-End QA | 70.45 | 51.84 | ✗ | - | LLM |
| Few-Shot QA | 71.54 | 52.56 | ✗ | - | LLM |
| Chain-of-Thought (Wei et al., 2022) | 65.37 | 53.48 | ✗ | - | LLM |
| Binder (Cheng et al., 2023) | 79.17 | 56.74 | ✓ | LLM + Program | Program |
| DATER (Ye et al., 2023) | 78.01 | 52.81 | ✓ | Program | LLM |
| CoTable (Wang et al., 2024) | **80.20** | **59.90** | ✓ | Program | LLM |
| Text-to-SQL (Rajkumar et al., 2022) | 64.71 | 52.90 | ✗ | Program | Program |
| LPA (Chen et al., 2020) | 68.90 | - | ✓ | Program | Program |
| POS (ours) | 78.31 | 54.80 | ✓ | Program | Program |
| GPT-4o-mini (`gpt-4o-mini-2024-07-18`) | | | | | |
| Binder (Cheng et al., 2023) | **84.63** | 58.86 | ✓ | LLM + Program | Program |
| DATER (Ye et al., 2023) | 80.98 | 58.83 | ✓ | Program | LLM |
| CoTable (Wang et al., 2024) | 84.24 | 55.60 | ✓ | Program | LLM |
| POS (ours) | 82.70 | **59.32** | ✓ | Program | Program |

Using GPT-4o-mini, POS reaches 82.70% on TabFact and **59.32%** on WikiTQ. While hybrid approaches like Binder and CoTable score higher on TabFact, POS achieves the **best** performance on WikiTQ. This demonstrates that with a more advanced language model, POS remains competitive while offering great interpretability, as shown in Sec. 4.1.3.

Table 7: Accuracy of POS across varying table sizes. The Pearson correlation reveals negligible relationships between table size and accuracy.

| | TabFact | | | WikiTQ | | |
| Size | Small | Medium | Large | Small | Medium | Large |
|---|---|---|---|---|---|---|
| Token Range | 30–109 | 109–188 | 188–804 | 135–638 | 638–1307 | 1307–33675 |
| Accuracy | 79.1% (533/674) | 85.2% (575/675) | 81.5% (550/675) | 56.1% (713/1448) | 46.7% (574/1448) | 47.8% (558/1448) |
| Correlation | | $-0.006$ | | | $-0.023$ | |

### 4.2.2 Performance vs. table size analysis

A natural concern for Table QA systems is whether the performance degrades as input tables grow larger and more complex. To answer this, we provide a quantitative evaluation of POS accuracy on TabFact and WikiTQ, stratified by table size. Specifically, following the methodology in Wang et al. (2024), we sort each table by token count (our measure for table size) and split datasets into three bins of equal size (small, medium, and large). We use GPT-4o-mini as the backbone LLM and report the QA accuracy in Tab. 7.

We observe that POS maintains stable accuracy on TabFact (79.1 → 85.2 → 81.5%, r=–0.006), though on WikiTQ performance drops from 56.1% (small) to 46.7% (medium) before recovering slightly to 47.8% (large) (<10 percentage points; r=–0.023). This suggests POS is resilient on some benchmarks but can suffer on tables of intermediate size in more complex tasks.

We find that POS is more robust than existing methods—such as Binder, DATER, and CoTable—which all suffer significant accuracy drops with large tables (i.e., tables longer than 4K tokens; see Table 3 in Wang et al. (2024)). For instance, Binder and DATER experience accuracy declines of 30–50 percentage points, dropping to 6.41% and 34.62%, respectively, while CoTable degrades to 44.87% on WikiTQ. This degradation is primarily due to the challenges LLMs face when processing long contexts. In contrast, POS's SQL-based executions remain resilient to variations in table length, making it well-suited for real-world scenarios where table sizes may scale to millions of tokens.

## 4.3 Efficiency Analysis

Table 8: Efficiency analysis on WikiTQ. **SC** denotes self-consistency usage, **LLM** represents the total number of LLM calls (detailed in **Breakdown**), and **DB** is the number of database queries. Notably, `POS` requires up to $25\times$ fewer calls/queries than other methods.

| Method | SC | LLM | Breakdown | DB |
|---|---|---|---|---|
| Binder (Cheng et al., 2023) | ✓ | 50 | GenerateSQL: 50 | 50 |
| DATER (Ye et al., 2023) | ✓ | 100 | Decompose: 40; Cloze: 20; GenerateSQL: 20; GenerateAnswer: 20 | 20 |
| CoTable (Wang et al., 2024) | ✓ | $\leq 25$ | Planning: $\leq 5$; GenerateArgs: $\leq 19$; GenerateAnswer: 1 | 5 |
| `POS` (ours) | ✗ | **4** | Planning: 2; GenerateSQL: 2 | **2** |

Efficiency in Table QA is crucial because reducing the number of LLM calls and database queries directly lowers computational costs and improves scalability. We leverage the efficiency benchmark introduced by Wang et al. (2024), which measures the number of LLM calls required to answer a WikiTQ question. Additionally, we propose using the number of database queries (i.e., table transformations) that reflects the computational workload on the table database.

As `POS` employs highly deterministic and atomic steps, it eliminates the need for the costly self-consistency prompting (Wang et al., 2022) required by Binder, DATER, and CoTable. As a result, `POS` requires only four LLM calls per question—significantly fewer than CoTable (25), Binder (50), and DATER (100). Regarding database queries, `POS` is also more efficient than others with only two queries per question, compared to five of CoTable, 50 of Binder, or 20 of DATER.

## 4.4 `POS` for Free-form Table QA

`POS` processes Table QA queries end-to-end using SQL commands, performing table transformations exclusively on the input table and without accessing to external knowledge. This SQL-only pipeline restricts `POS`'s applicability to tasks requiring creativity, such as generating paragraph-like answers. To address this, we extend `POS` to the free-form Table QA task (FeTaQA) (Nan et al., 2022) by integrating an LLM call in the final step to generate free-form natural language answers, following the algorithm illustrated in Fig. 2.

Table 9: Results on FeTaQA, a free-form Table QA task (using GPT-4o-mini as the base LLM). `POS` outperforms both 0-shot end-to-end and few-shot QA.

| Method | BLEU | ROUGE-1 | ROUGE-L |
|---|---|---|---|
| End-to-End QA | 18.99 | 51.92 | 46.44 |
| Few-Shot QA | 19.18 | 53.32 | 46.86 |
| POS | **20.16** | **54.70** | **48.69** |

We compare `POS` with End-to-End and Few-Shot QA and find that `POS` consistently outperforms both methods (see Tab. 9). This improved performance (+0.98 points in BLEU and +1.83 points in ROUGE-L compared to Few-Shot QA) is attributed to intermediate SQL executions of `POS`, which retrieve fine-grained and relevant information to generate final answers. In contrast, End-to-End and Few-Shot QA process the entire input table at once, making it challenging for the model to pinpoint and make use of the correct data.

Please note that we compare `POS` against these two LLM-only baselines—selected specifically to showcase the adaptability of `POS` in generating free-form natural language responses while preserving its interpretability advantages (in intermediate table transformations). As we prioritize interpretability, we have not optimized `POS` accuracy in free-form Table QA, and therefore we do not include many hybrid QA baselines in Tab. 9.

## 5  Discussion and Future Works

We address pertinent questions regarding the robustness and interpretability of `POS`; more are in Appendix C.

**How does `POS` handle real-world "messy" tables?**  We have so far evaluated on benchmark tables that are semi-structured and relatively clean (as preprocessed by Wang et al. (2024) and Ye et al. (2023)). In practice, tables "in the wild" may have merged cells, nested headers, irregular formats, or other noise. `POS` in its current form does not explicitly tackle such noise. A practical extension would be to integrate a preprocessing step that normalizes and restructures messy tables (e.g., using a tool like NormTab (Nahid & Rafiei, 2024a)) before applying `POS`. This could allow `POS` to maintain high accuracy and interpretability even on more complex, non-canonical tables.

**How can `POS` detect errors in its own SQL generation?** `POS` uses a fallback that could diminish the program-only nature: if SQL execution fails, an LLM answers the question end-to-end—triggered in 3.16% of TabFact and 13.58% of WikiTQ samples, similar to other Table QA methods (Ye et al., 2023; Cheng et al., 2023; Wang et al., 2024). Tab. 6 reports `POS` accuracy using this fallback mechanism. To minimize such fallbacks, we propose implementing a proactive error detection and correction mechanism that validates each generated SQL query against the table schema. When an error is detected, the model regenerates the query using the execution history log as context.

**Is `POS` completely interpretable?**  No; no AI system is entirely interpretable. In computer vision, for example, methods such as concept bottleneck models provide human-understandable concepts (Ismail et al., 2023) for model decisions, but the reasoning underlying the network to generate the concepts remains unintelligible.  Likewise, LLMs can generate chain-of-thought rationales, but the internal hidden states driving these thoughts cannot be directly understood (Wei et al., 2022; Madsen et al., 2024). `POS` bottlenecks the decision-making into human-interpretable operations—namely, natural-language steps that are translated into SQLs.  They deterministically produce the answer, allowing users to follow the logical chain of table transformations.  Although the underlying LLM may involve black-box internal processes (in generating natural-language steps), our focus is on making model's decision-making interpretable, aligning with prior and current interpretability research (Taesiri et al., 2022; Ismail et al., 2023; Zhao et al., 2024).

**Can every table-based question be decomposed into a set of atomic steps?**  Yes; by design `POS` breaks any question over a table into a sequence of *atomic* operations—filter, select, group, aggregate, sort, limit, join, set operations, and so on—each implemented by a single, standalone SQL statement.  Because SQL is relationally complete (i.e., it can express every relational-algebra query), this set of primitives (or atomic operations) can express *any* query, from simple counts to complex multi-table analytics[2].

The only limitation arises for truly *free-form* queries that demand narrative or summary generation (e.g., "Summarize the basic information of the football clubs in Saint Petersburg."). In such cases, `POS` still applies atomic SQL steps to extract and prepare the data, then defers only the final *writing* step to a language model, preserving full transparency of all intermediate transformations.

**What are the most common failure modes of `POS`?** We observed that errors in `POS` mostly originate from the planning stage rather than from the Step-to-SQL. For example, the planner often omits necessary condition checks in its atomic steps or performs steps in the orders that lead to the wrong answers (refer to Appendix H.1.3 for the importance of step order). As LLMs continue to improve their planning capabilities (Huang et al., 2022), we expect `POS` to benefit accordingly in terms of performance while retaining its strong interpretability. Please refer to Appendix J for examples and analysis of `POS` errors.

**Why `POS` explanations do not contain SQLs?**  We designed `POS` to present natural-language steps rather than raw SQL because, in our user studies, including SQL in the explanations tended to overwhelm users.  The natural language descriptions are equivalent to the SQL commands—thanks to the atomicity constraint, which guarantees minimal-to-no discrepancy between the two—yet they are far more readable and accessible. This design choice strikes a balance between interpretability and clarity, ensuring that users grasp the model's reasoning without being burdened by technical SQL syntax.

---

[2]https://www.cwblogs.com/posts/relational-algebra/

# 6 Conclusion

We introduce Plan-of-SQLs (`POS`), an interpretable, effective, and efficient approach to Table QA with large language models. `POS` decomposes a table question into simple atomic steps and executes each step using SQL, thereby ensuring that each transformation in the reasoning process is transparent. This design fills a critical gap in current LLM-based Table QA models, which often produce answers without clear explanations. Our experiments demonstrate that prioritizing interpretability does not come at the expense of performance: `POS` achieves explanation quality superior to existing methods (as confirmed by both human evaluations and LLM judges) and competitive accuracy on Table QA benchmarks. Moreover, `POS` accomplishes this with an order-of-magnitude fewer LLM calls and database queries than prior approaches, making it a more efficient and scalable solution for real-world use. We also find that human evaluators and LLM judges largely agree on their evaluations of explanations, indicating that LLM-based automated evaluation is feasible. The only significant limitation we observe is rooted in the LLM's planning capability—if the LLM produces a suboptimal plan, `POS` can err, which aligns with recent observations by Zhao et al. (2024). Encouragingly, as LLMs become more powerful in planning, we expect the performance of `POS` to naturally improve while preserving its inherent interpretability. In summary, `POS` represents a step toward Table QA models that not only deliver accurate answers but also provide human-understandable explanations.

## Acknowledgement

GN was supported by an Auburn University Presidential Graduate Research Fellowship. AN was supported by NSF Grant #2145767, NAIRR award #240116, donations from NaphCare Foundation, and research gifts from Adobe Research.

## Disclaimer

This paper was prepared for informational purposes by the Artificial Intelligence Research group of JPMorgan Chase & Co. and its affiliates ("JP Morgan") and is not a product of the Research Department of JP Morgan. JP Morgan makes no representation and warranty whatsoever and disclaims all liability, for the completeness, accuracy or reliability of the information contained herein. This document is not intended as investment research or investment advice, or a recommendation, offer or solicitation for the purchase or sale of any security, financial instrument, financial product or service, or to be used in any way for evaluating the merits of participating in any transaction, and shall not constitute a solicitation under any jurisdiction or to any person, if such solicitation under such jurisdiction or to such person would be unlawful.

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

# Appendix for:
# Interpretable LLM-based Table Question Answering

## A  Explanation Methods for Table QA

In this section, we present visual explanations for Table QA models, which help bridge the gap between model behaviors and human understanding. Each visualization provides insights into how the model leverages the input table, highlighting the key information used in its reasoning process.

For our experiments, we use TabFact (Chen et al., 2020) and WikiTQ (Pasupat & Liang, 2015), running each method on the test sets of 2,024 and 4344 samples. `POS`, CoTable, and DATER all use the same visualization format described in  Sec. 3.3.

We use four different methods for explaining Table QA answers: Text-to-SQL, DATER, CoTable, and `POS` (ours). Each method offers a unique set of information as follows:

- Text-to-SQL (Rajkumar et al., 2022). This method translates the question into a single SQL command, which is then executed on the input table to produce an answer. *Explanation Generation:* We present the *generated SQL command* along with the question and the input table to show how the final answer is derived. Although the SQL itself can be clear to experts, it may require additional domain knowledge to interpret (see Fig. 4 for an example). Therefore, the we recruited Computer Science students (who have SQL expertise) for our human study.

- DATER (Ye et al., 2023). DATER solves a natural language question via extracting a relevant subtable and uses SQLs to verify partial facts (Fig. 5). *Explanation Generation:* We extract (1) the *subtable* selected by DATER from the input table, (2) the *verified facts*, and (3) *attribution maps* that highlight which table cells DATER considers relevant. Furthermore, we show step descriptions (e.g., "Select rows") from DATER's working logs, allowing users to track subtable extraction (see Fig. 5).

- CoTable (Wang et al., 2024). CoTable processes queries by planning a sequence of abstract function calls (e.g., `f_sort`, `f_select_row`, etc.), each responsible for table transformation. *Explanation Generation:* Similar to `POS`, we visualize each intermediate table along with *attribution maps*, but the transformations are represented by function names and their arguments (e.g., `f_select_row(1)`). Although this step-by-step approach can be informative, arguments are not well-justified and the final answer relies on LLM-driven "black-box" reasoning (see Fig. 6).

- `POS` (Ours). `POS` decomposes a natural language question into a sequence of *atomic* reasoning steps, where each step is explicitly translated into an executable SQL command. *Explanation Generation:* For `POS`, we present the natural language description of each step—these are functionally equivalent to the underlying SQL commands, but far more accessible and readable to users. Users can inspect each transformation through the clear and concise natural language steps, supported by attribution maps highlighting the relevant table cells for every operation (see Fig. 7).

**Comparison of DATER and CoTable.** Empirically, we find no major difference in their overall "informativeness" (Sec. 4.1.3), although CoTable displays intermediate tables more explicitly. Both methods are still considered **hybrid**—they rely on partial SQLs or function calls combined with LLM-driven reasoning. For tasks demanding deeper inspection of intermediate steps (e.g., verification complex filters), CoTable's step-by-step interface may be more revealing than DATER's subtable-based approach.

**In general,** by generating method-specific explanations under a unified visualization format, we can fairly compare how each Table QA methods explain reasoning and arrives at its final answer. This shared framework allows us to examine strengths, weaknesses, and interpretability trade-offs across different approaches in earlier sections.

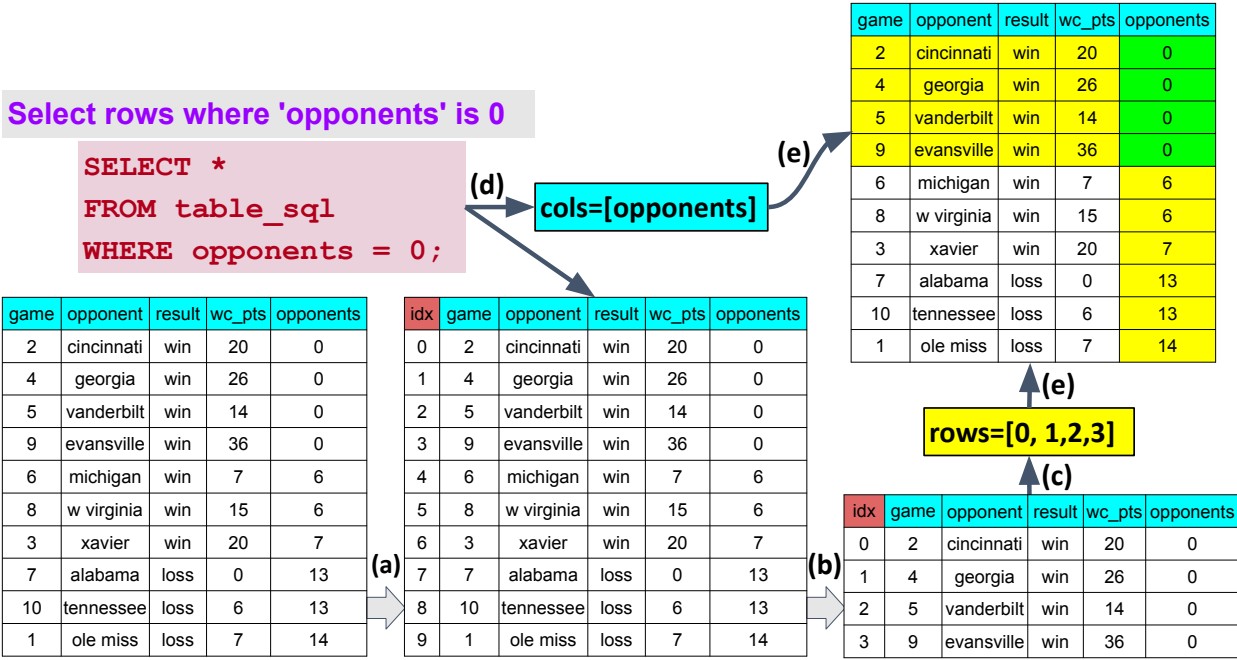

Figure 3: Generating attributions maps for `POS`. Column idx is added to track row attribution.

## A.1 Attribution Maps

During the execution of each SQL command, we perform the following steps:

- **Adding the tracking index column:** Before executing an SQL, we add a tracking index column to the current table. This column contains the original row indices from the initial table— Fig. 3**(a)**.

- **Executing the SQL command:** An SQL command is executed on the table with the tracking index column, producing a modified table— Fig. 3**(b)**.

- **Identifying selected rows:** After execution, we use the tracking index column to identify which rows have been selected or filtered by the SQL command— Fig. 3**(c)**.

- **Identifying selected columns:** We parse the SQL command to extract the columns involved in the operation— Fig. 3**(d)** (see Appendix E).

- **Visualizing an attribution map:** The prior information allow us to generate an attribution map for the intermediate tables— Fig. 3**(e)**. The index column is also removed at this step.

Since both rows and columns can be attributed within an operation, `POS` offers a *distinct advantage* over previous works (Ye et al., 2023; Wang et al., 2024)—accurately attributing responsible cells for each transformation. For example, when an SQL command includes a condition that requires a cell to match a specific value or range (e.g., `WHERE opponents = 0`), we can determine which cells in the `opponents` column satisfy this condition and are thus responsible for the answer— Fig. 3**(e)**.

## A.2 Explanations as Chains of Attribution Maps

At each step of the table transformation process, we visualize attribution maps on the input table for that step, highlighting the data selected or filtered in the current operation. Rows and columns that contain relevant data for the operation are yellow-highlighted, while cells that match the specific condition in this step are green-highlighted.

Using the information obtained from the plan execution and attribution maps, we combine the three components: (1) intermediate tables; (2) attribution maps; and (3) step description; to create an explanation.

We present the explanation in a *chain of attribution maps*, helping users visually follow the sequence of transformations and understand how each table cell contributes to the final answer. We show representative examples of explanation methods in our study below.

---

**Statement: the wildcats kept the opposing team scoreless in four games**

**Input Table: 1947 kentucky wildcats football team**

| game | date | opponent | result | wildcats_points | opponents | record |
|------|------|----------|--------|-----------------|-----------|--------|
| 1 | 9999-09-20 | ole miss | loss | 7 | 14 | 0 - 1 |
| 2 | 9999-09-27 | cincinnati | win | 20 | 0 | 1 - 1 |
| 3 | 9999-10-04 | xavier | win | 20 | 7 | 2 - 1 |
| 4 | 9999-10-11 | 9 georgia | win | 26 | 0 | 3 - 1 , 20 |
| 5 | 9999-10-18 | 10 vanderbilt | win | 14 | 0 | 4 - 1 , 14 |
| 6 | 9999-10-25 | michigan state | win | 7 | 6 | 5 - 1 , 13 |
| 7 | 9999-11-01 | 18 alabama | loss | 0 | 13 | 5 - 2 |
| 8 | 9999-11-08 | west virginia | win | 15 | 6 | 6 - 2 |
| 9 | 9999-11-15 | evansville | win | 36 | 0 | 7 - 2 |
| 10 | 9999-11-22 | tennessee | loss | 6 | 13 | 7 - 3 |

**SQL Command:**

```
SELECT
    CASE
        WHEN COUNT(*) = 4 THEN 'TRUE'
        ELSE 'FALSE'
    END
FROM table_sql
WHERE opponents = 0;
```

Figure 4: **Text-to-SQL** explanations provide only the SQL command, which is intuitive for SQL users.

**Statement: the wildcats kept the opposing team scoreless in four games in the table: the wildcats kept the opposing team scoreless in 4 games.**

**Input Table: 1947 kentucky wildcats football team**

> Step 1: Select Rows (row 4, row 5, row 3, row 2, row 9) and Select Columns (opponents, wildcats points, game)

| game | date | opponent | result | wildcats points | opponents | record |
|---|---|---|---|---|---|---|
| 1 | sept 20 | ole miss | loss | 7 | 14 | 0 - 1 |
| 2 | sept 27 | cincinnati | win | 20 | 0 | 1 - 1 |
| 3 | oct 4 | xavier | win | 20 | 7 | 2 - 1 |
| 4 | oct 11 | 9 georgia | win | 26 | 0 | 3 - 1 , 20 |
| 5 | oct 18 | 10 vanderbilt | win | 14 | 0 | 4 - 1 , 14 |
| 6 | oct 25 | michigan state | win | 7 | 6 | 5 - 1 , 13 |
| 7 | nov 1 | 18 alabama | loss | 0 | 13 | 5 - 2 |
| 8 | nov 8 | west virginia | win | 15 | 6 | 6 - 2 |
| 9 | nov 15 | evansville | win | 36 | 0 | 7 - 2 |
| 10 | nov 22 | tennessee | loss | 6 | 13 | 7 - 3 |

> Sub-table Selection

| opponents | wildcats points | game |
|---|---|---|
| 0 | 20 | 2 |
| 7 | 20 | 3 |
| 0 | 26 | 4 |
| 0 | 14 | 5 |
| 0 | 36 | 9 |

> Contextual information: the wildcats kept the opposing team scoreless in 4 games.

## Prompting LLM for the final answer... >>>

**Prediction: TRUE**

Figure 5: **DATER** explanations contain sub-table selection, contextual information (or verified facts), and highlights that reveal which input features influence the final answer.

**Statement: the wildcats kept the opposing team scoreless in four games**

**Input Table: 1947 kentucky wildcats football team**

Step 1: f_select_row(row 1, row 2, row 3, row 4, row 8)

| game | date | opponent | result | wildcats points | opponents | record |
|------|------|----------|--------|-----------------|-----------|--------|
| 1 | sept 20 | ole miss | loss | 7 | 14 | 0 - 1 |
| 2 | sept 27 | cincinnati | win | 20 | 0 | 1 - 1 |
| 3 | oct 4 | xavier | win | 20 | 7 | 2 - 1 |
| 4 | oct 11 | 9 georgia | win | 26 | 0 | 3 - 1 , 20 |
| 5 | oct 18 | 10 vanderbilt | win | 14 | 0 | 4 - 1 , 14 |
| 6 | oct 25 | michigan state | win | 7 | 6 | 5 - 1 , 13 |
| 7 | nov 1 | 18 alabama | loss | 0 | 13 | 5 - 2 |
| 8 | nov 8 | west virginia | win | 15 | 6 | 6 - 2 |
| 9 | nov 15 | evansville | win | 36 | 0 | 7 - 2 |
| 10 | nov 22 | tennessee | loss | 6 | 13 | 7 - 3 |

Step 2: f_select_column(game, wildcats points, opponents)

| game | date | opponent | result | wildcats points | opponents | record |
|------|------|----------|--------|-----------------|-----------|--------|
| 2 | sept 27 | cincinnati | win | 20 | 0 | 1 - 1 |
| 3 | oct 4 | xavier | win | 20 | 7 | 2 - 1 |
| 4 | oct 11 | 9 georgia | win | 26 | 0 | 3 - 1 , 20 |
| 5 | oct 18 | 10 vanderbilt | win | 14 | 0 | 4 - 1 , 14 |
| 9 | nov 15 | evansville | win | 36 | 0 | 7 - 2 |

Step 3: f_sort_column(opponents)

| game | wildcats points | opponents |
|------|-----------------|-----------|
| 2 | 20 | 0 |
| 3 | 20 | 7 |
| 4 | 26 | 0 |
| 5 | 14 | 0 |
| 9 | 36 | 0 |

Step 4: simple_query()

| game | wildcats points | opponents |
|------|-----------------|-----------|
| 2 | 20 | 0 |
| 4 | 26 | 0 |
| 5 | 14 | 0 |
| 9 | 36 | 0 |
| 3 | 20 | 7 |

**Prompting LLM for the final answer... >>>**

**Prediction: TRUE**

Figure 6: **CoTable** explanations present intermediate tables and highlights, showing key steps in data transformation. Additionally, the steps are presented through function names and their arguments.

**Statement: the wildcats kept the opposing team scoreless in four games**

**Input Table: 1947 kentucky wildcats football team**

Step 1: Order the table by 'opponents' in ascending order.

| game | date | opponent | result | wildcats_points | opponents | record |
|---|---|---|---|---|---|---|
| 1 | 9999-09-20 | ole miss | loss | 7 | 14 | 0 - 1 |
| 2 | 9999-09-27 | cincinnati | win | 20 | 0 | 1 - 1 |
| 3 | 9999-10-04 | xavier | win | 20 | 7 | 2 - 1 |
| 4 | 9999-10-11 | 9 georgia | win | 26 | 0 | 3 - 1 , 20 |
| 5 | 9999-10-18 | 10 vanderbilt | win | 14 | 0 | 4 - 1 , 14 |
| 6 | 9999-10-25 | michigan state | win | 7 | 6 | 5 - 1 , 13 |
| 7 | 9999-11-01 | 18 alabama | loss | 0 | 13 | 5 - 2 |
| 8 | 9999-11-08 | west virginia | win | 15 | 6 | 6 - 2 |
| 9 | 9999-11-15 | evansville | win | 36 | 0 | 7 - 2 |
| 10 | 9999-11-22 | tennessee | loss | 6 | 13 | 7 - 3 |

Step 2: Select rows where 'opponents' is 0.

| game | date | opponent | result | wildcats_points | opponents | record |
|---|---|---|---|---|---|---|
| 2 | 9999-09-27 | cincinnati | win | 20 | 0 | 1 - 1 |
| 4 | 9999-10-11 | 9 georgia | win | 26 | 0 | 3 - 1 , 20 |
| 5 | 9999-10-18 | 10 vanderbilt | win | 14 | 0 | 4 - 1 , 14 |
| 9 | 9999-11-15 | evansville | win | 36 | 0 | 7 - 2 |
| 6 | 9999-10-25 | michigan state | win | 7 | 6 | 5 - 1 , 13 |
| 8 | 9999-11-08 | west virginia | win | 15 | 6 | 6 - 2 |
| 3 | 9999-10-04 | xavier | win | 20 | 7 | 2 - 1 |
| 7 | 9999-11-01 | 18 alabama | loss | 0 | 13 | 5 - 2 |
| 10 | 9999-11-22 | tennessee | loss | 6 | 13 | 7 - 3 |
| 1 | 9999-09-20 | ole miss | loss | 7 | 14 | 0 - 1 |

Step 3: Use a `CASE` statement to return TRUE if the number of rows is equal to 4, otherwise return FALSE.

| game | date | opponent | result | wildcats_points | opponents | record |
|---|---|---|---|---|---|---|
| 2 | 9999-09-27 | cincinnati | win | 20 | 0 | 1 - 1 |
| 4 | 9999-10-11 | 9 georgia | win | 26 | 0 | 3 - 1 , 20 |
| 5 | 9999-10-18 | 10 vanderbilt | win | 14 | 0 | 4 - 1 , 14 |
| 9 | 9999-11-15 | evansville | win | 36 | 0 | 7 - 2 |

| verification_result |
|---|
| TRUE |

**Prediction: TRUE**

Figure 7: `POS` explanations contain intermediate tables and highlights. The green-highlighted cells indicate where the information in the table matches the conditions specified in the natural language steps.

### A.3 Self-Explanation of Table QA Models

An interesting question is whether a simple, post-hoc explanation method might achieve similar interpretability. To explore this, we adopt a widely used "self-explanation" baseline (Madsen et al., 2024; Chen et al., 2023b; Agarwal et al., 2024). Specifically, the LLM is first prompted to produce an answer and then asked to retrospectively explain how it arrived at that answer (see Fig. 8). We compare this post-hoc approach with our proposed `POS` and other XAI baselines (Text-to-SQL, DATER, and CoTable) in Tab. 10, focusing on three XAI benchmarks in the TabFact dataset using `GPT-4o-mini`.

Table 10: Comparison of post-hoc self-explanation vs. other explanation methods. Lower is better for *Preference*; higher is better for *Forward Simulation* and *Verification*.

| Method | Preference ($\downarrow$) | Forward Sim. ($\uparrow$) | Verification ($\uparrow$) |
|---|---|---|---|
| Self-explanation | 5.00 | 65.98% | 68.03% |
| Text-to-SQL | 3.99 | 65.67% | 55.37% |
| DATER | 2.71 | 73.57% | 55.43% |
| CoTable | 1.77 | 76.53% | 61.36% |
| **POS** (Ours) | **1.53** | **81.61%** | **76.74%** |

Our main finding is that post-hoc self-explanation outperforms Text-to-SQL (always the worst) but remains significantly less effective than our proposed `POS` method. This may be attributed to its lack of faithfulness, as extensively identified in previous studies (Chen et al., 2023b; Madsen et al., 2024; Agarwal et al., 2024), where the explanation often fails to align with the model's true reasoning process. In contrast, `POS` explanations are generated and *executed* at each step, inherently tying them to the actual transformations applied to the data. This design, as demonstrated in our earlier ablation and user studies, results in greater understanding and more actionable interpretability for downstream tasks like verification and forward simulation. We show an example of self-explanation below.

**Prompt:**
*You are given a Statement and an Input Table.*

*Your Task:*
*1. Verify if the Statement if True or False agaisnt the Input Table.*
*2. Explain your verification process based on the provided info.*
*3. Your answer must be TRUE or FALSE only.*

**Statement:** the wildcats kept the opposing team scoreless in four games
**Input Table Caption:** 1947 kentucky wildcats football team

| game | date | opponent | result | wildcats_points | opponents | record |
|------|------|----------|--------|-----------------|-----------|--------|
| 1 | 9999-09-20 | ole miss | loss | 7 | 14 | 0 - 1 |
| 2 | 9999-09-27 | cincinnati | win | 20 | 0 | 1 - 1 |
| 3 | 9999-10-04 | xavier | win | 20 | 7 | 2 - 1 |
| 4 | 9999-10-11 | 9 georgia | win | 26 | 0 | 3 - 1, 20 |
| 5 | 9999-10-18 | 10 vanderbilt | win | 14 | 0 | 4 - 1, 14 |
| 6 | 9999-10-25 | michigan state | win | 7 | 6 | 5 - 1, 13 |
| 7 | 9999-11-01 | 18 alabama | loss | 0 | 13 | 5 - 2 |
| 8 | 9999-11-08 | west virginia | win | 15 | 6 | 6 - 2 |
| 9 | 9999-11-15 | evansville | win | 36 | 0 | 7 - 2 |
| 10 | 9999-11-22 | tennessee | loss | 6 | 13 | 7 - 3 |

**LLM Self-explanation:**
**Verification:** TRUE

*To verify the statement "the wildcats kept the opposing team scoreless in four games", we need to examine the input table, which lists the results of the Kentucky Wildcats football games. The key information here is that the Wildcats are considered to have kept the opposing team scoreless if the column "opponents_points" shows a value of "0" for that game.*
*. . .*

*We can go through the table and count how many games the Wildcats kept the opposing team scoreless (opponents_points = 0):*
*Game 1: Ole Miss – Opponents' points = 14 (not scoreless)*
*Game 2: Cincinnati – Opponents' points = 0 (scoreless)*
*Game 3: Xavier – Opponents' points = 7 (not scoreless)*
*Game 4: Georgia – Opponents' points = 0 (scoreless)*
*Game 5: Vanderbilt – Opponents' points = 0 (scoreless)*
*Game 6: Michigan State – Opponents' points = 6 (not scoreless)*
*Game 7: Alabama – Opponents' points = 13 (not scoreless)*
*Game 8: West Virginia – Opponents' points = 6 (not scoreless)*
*Game 9: Evansville – Opponents' points = 0 (scoreless)*
*Game 10: Tennessee – Opponents' points = 13 (not scoreless)*
*From the table, we see that the Wildcats kept the opposing team scoreless in four games (games 2, 4, 5, and 9).*

Figure 8: An example of a post-hoc self-explanation. The LLM first produces an answer (TRUE) and then justifies it by referencing the table entries that support the claim.

## B  Baselines and Related Works

**LLM-only.**   These approaches rely solely on LLMs to generate answers without explicitly performing table transformations. End-to-End QA prompts the LLM to generate answers directly from the input table and question. Similarly, Few-Shot QA (Brown et al., 2020) includes few-shot examples $(T, Q, A)$ as the context to aid the LLM. In contrast, Chain-of-Thought (Wei et al., 2022) prompts the LLM to explain its reasoning process step-by-step before delivering the final answer.

**Program-only.**   Program-based approaches generate explicit programs to perform table transformation and answer the question. Latent Program Algorithm (LPA) (Chen et al., 2020) frames TabFact verification as a program synthesis task, converting input queries into sequential operations (e.g., min, max, count, filter) executed via Python-Pandas. On the other hand, Text-to-SQL (Rajkumar et al., 2022) translates a natural language question directly into a single SQL command, which is then applied to the input table to generate the answer.

**Hybrid.**   Hybrid approaches combine the strengths of LLM reasoning and programs to perform Table QA and achieve state-of-the-art accuracy. DATER (Ye et al., 2023) uses an LLM to extract relevant sub-tables, while breaking queries into sub-queries and executing SQL commands to retrieve factual information. Binder (Cheng et al., 2023) takes a different approach by converting natural language questions into executable programs. It blends API calls with symbolic language interpreters like SQL or Python to address reasoning gaps that cannot be handled through offline methods alone. Lastly, CoTable (Wang et al., 2024) dynamically plans a sequence of predefined table operations–such as selecting rows or adding columns, allowing it to iteratively transform the table based on the intermediate information. Despite their differences, they all share a common strategy: they input the final simplified table along with the original question into an LLM to produce the final answer.

**Program-aided language models vs. Plan-of-SQLs.**   Although both PAL (Gao et al., 2023) and `POS` break questions into executable, program-based steps, they differ fundamentally in focus. PAL targets general math and symbolic problems by emitting a single Python script—comments and code intertwined—making it hard for non-programmers to trace each operation. In contrast, `POS` is built specifically for Table QA: it forces every reasoning step to be "atomic" (one condition and one variable at a time), translating each into a trivial SQL query. This guarantees that users can inspect and verify each intermediate result before moving on. Moreover, `POS` produces fine-grained attribution maps over the actual tables at every step, so users see exactly which rows and columns were involved—whereas PAL's logic remains buried in Python code, opaque to those without coding expertise.

## C  More Ablation, Discussion, and Future Works

**What is the impact of removing SQL execution from `POS`?** We investigate this impact in Sec. 4.1.6. Removing SQL execution not only diminishes interpretability, but also affects accuracy on benchmark datasets. In this ablation, the LLM is tasked with directly transforming the table rather than executing SQL commands. While this leads to a negligible decrease in Table QA accuracy on TabFact, it causes a substantial drop on WikiTQ. This discrepancy suggests that relying solely on the black-box reasoning of LLMs for table transformations can severely impact model accuracy—likely due to LLM hallucinations or errors when handling complex tables (Chen, 2023; Wang et al., 2024). Moreover, bypassing SQL removes a layer of transparency, as the table transformations are no longer traceable, a reduction in interpretability that is further quantified by Tab. 5.

**Have we evaluated explanation effectiveness in free-form Table QA?** No; our work has not yet investigated the interpretability in free-form Table QA (e.g., FeTaQA), where the final answer is generated through a black-box operation of LLMs—making `POS` explanations no longer 100% faithful or comprehensive. A promising explanation approach is to develop hybrid explanations that blend `POS`-generated **intermediate-step** explanations with groundings (Hendricks et al., 2018) for the final answer. In this hybrid format, intermediate steps generated by `POS` provide a transparent view into the intermediate reasoning process while grounding algorithms are employed to anchor the final response to the intermediate tables.

**What tabular questions are not SQL-decomposable?** There are questions that cannot be broken down into Plans of SQLs (or sequential SQL queries). To our knowledge, they are:

- **Free-form narrative queries:** e.g., "Summarize the basic information of the football clubs in Saint Petersburg". They require surface-level composition or paraphrasing, not a relational result set.

- **World-knowledge or multi-table reasoning outside the given database:** e.g., "Which of these universities is the oldest in the country"? Suppose that the table omits founding years. They require lookup beyond the input table; `POS` atomic SQL over the local table is, by construction, blind to external facts.

To automatically detect questions that are not suitable for `POS`, there are simple auto-detection strategies:

- **Rule-based heuristics**: This is the most naive solution. We flag questions containing key verbs such as "summarize", "describe", or "compare" as likely free-form or analytic.

- **Execution-failure signals**: Run the Step-to-SQL execution; if step converts to invalid SQL (Step-to-SQL raises error), flag as non-decomposable. This uses the same execution-validation hooks we have already described for error handling in Sec. 5.

- **LLM-based classification**: We can prompt the NL planner to emit a Boolean "SQL-decomposable?" tag or fine-tune a lightweight classifier on a small labeled dataset of decomposable vs. non-decomposable questions.

**SQL or Python; which is better for Table QA?**

To see the difference, consider the query "top 3 departments by average salary among active employees over 40". A Python solution might look like:

```
data = load_table("employees.csv")
step1 = [r for r in data if r["age"] > 40]
step2 = [r for r in step1 if r["status"] == "active"]
# custom code to group by department and compute averages
# then sort and slice the top 3
```

Using Python here has a few drawbacks:

- **Memory & speed:** We must load the entire table into Python and loop over every row, which is slow and memory-intensive for large datasets.

- **Custom logic:** Grouping, aggregating, and sorting require handwritten code (counters, loops, temporary structures) instead of built-in operations.

- **No indexing:** Python loops cannot leverage database indexes or optimized query planners, leading to poor scalability.

SQL is more preferable for Table QA because:

- **Declarative clarity:** We specify *what* we want (e.g., "Select rows where age > 40"), instead of using 3 steps sequentially: loop, filter > 40, and put to the list. This makes program more concise, self-documenting, and readable than using Python.

- **Engine optimizations:** Modern database engines have extensively optimized query planning, and indexing; they can process large tables far faster and more memory-efficiently than Python loops[3].

- **Consistency:** SQL's syntax and semantics are stable across databases (e.g., Oracle, MySQL, Microsoft SQL Server, and PostgreSQL), so the same query runs everywhere. Python script, by contrast, require custom connectors, custom layers, and may vary with library versions.

---

[3]https://airbyte.com/blog/sql-vs-python-data-analysis

**Ablation study on POS accuracy**

In addition to the ablation study we conducted on interpretability (in Sec. 4.1.6), we run another ablation study on the accuracy of POS on both TabFact and WikiTQ to pinpoint modules which may not be necessary for either interpretability or performance.

Table 11: Accuracy (%) on TabFact and WikiTQ using GPT-4o-mini.

| Variant | TabFact (%) | WikiTQ (%) |
|---|---|---|
| POS (Full) | 82.70 | 59.32 |
| – Atomic operations | 78.11 (–4.59) | 50.02 (–9.30) |
| – NL Planning | 77.90 (–4.80) | 48.27 (–11.05) |
| – SQL execution | 79.84 (–2.86) | 27.05 (–32.37) |

**We find that:**

- Replacing POS's SQL transformations with end-to-end LLM calls causes the biggest accuracy loss—especially on WikiTQ (59.32 → 27.05%, –32.37 pp) and a smaller drop on TabFact (82.70 → 79.84%, –2.86 pp).

- Removing either the natural-language planning or atomic-step enforcement each costs approximately 4–5 pp on TabFact and 9–11 pp on WikiTQ.

While all three components are important, robust SQL execution has the largest impact on accuracy, followed by step-by-step NL planning, and then atomic decomposition.

# D  Qualitative Examples for `POS` Explanations without Atomicity

In this section, we provide qualitative examples of `POS` explanations *with* and *without* atomicity enforcement in NL Planning. Removing atomicity from the plan steps can negatively impact interpretability, as the added complexity makes it harder for users to understand the model's reasoning process.

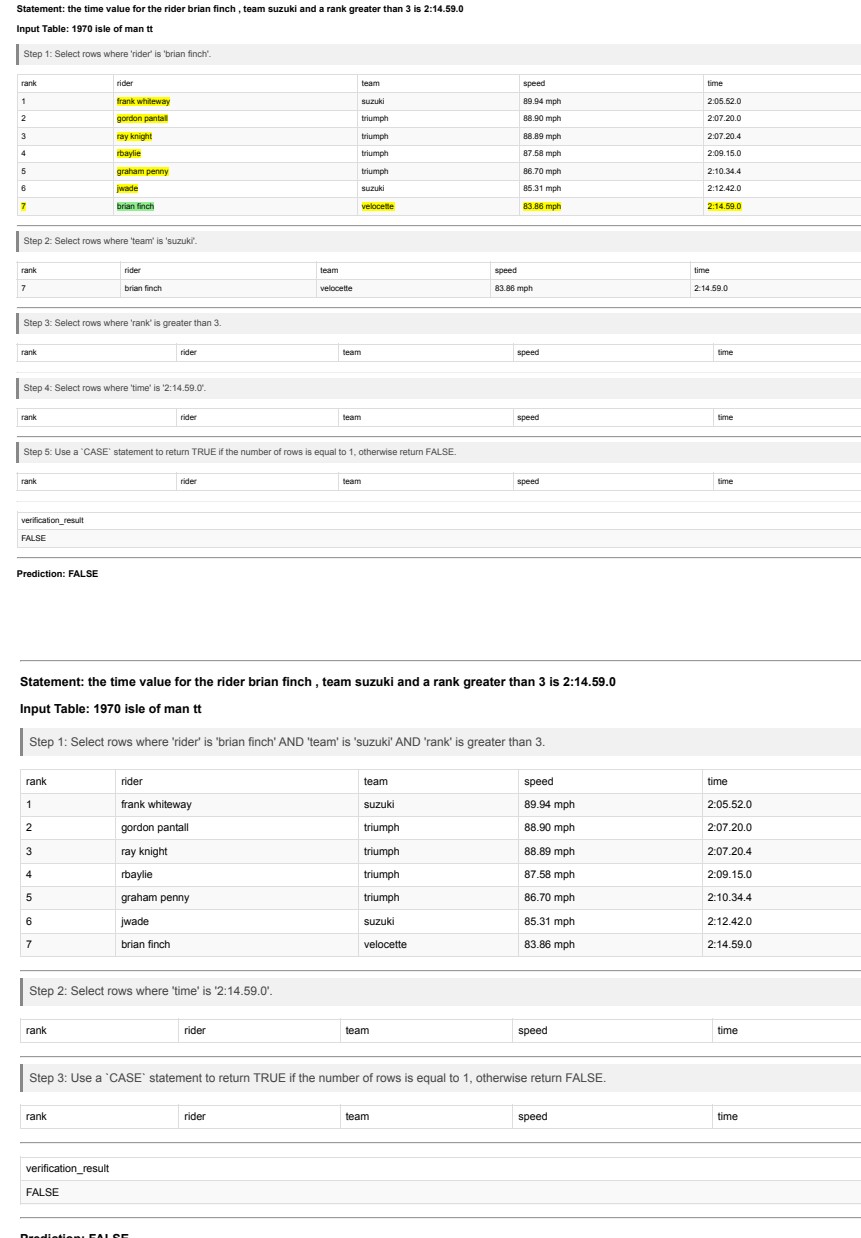

Figure 9: Upper: `POS` explanation with atomicity. Lower: `POS` explanation without atomicity.

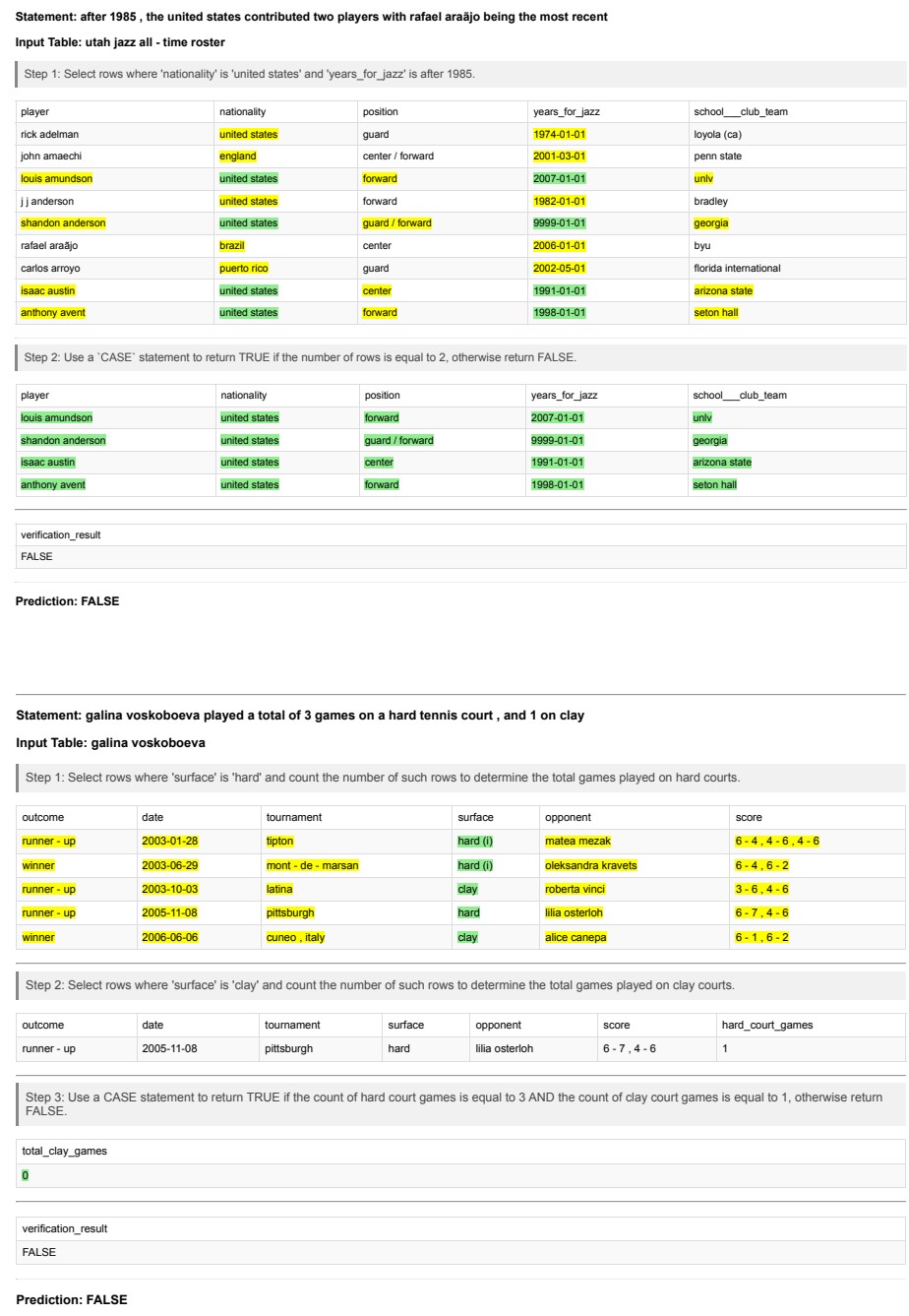

Figure 10: Two `POS` explanations without atomicity. The steps are compound and the attribution maps are non-trivial to comprehend.

# E Extracting Columns from SQL Commands

In this section, we detail the algorithm to analyze SQL queries and identify the columns used within them.

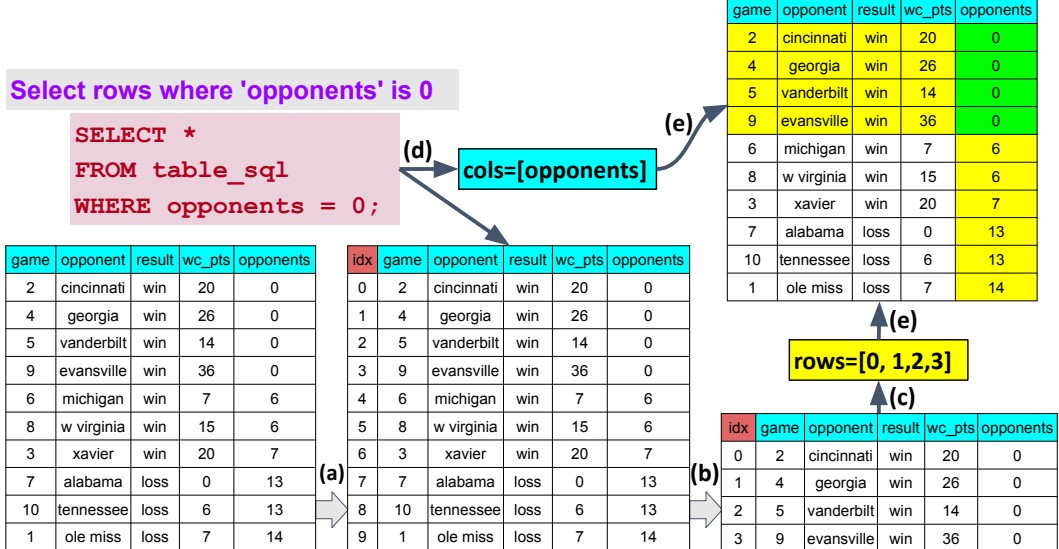

Figure 11: Data-attribution tracking algorithm for `POS`.

## E.1 Algorithm Overview

The algorithm follows these main steps:

1. **Preprocessing:** Remove comments and normalize whitespace in the SQL query.

2. **Column Extraction:** Parse different clauses of the SQL query to identify column names:
   - `SELECT` clause: Extract both regular columns and those used in functions.
   - `WHERE` clause: Identify columns used in conditions.
   - `ORDER BY` clause: Extract columns used for sorting.

3. **Filtering:** Compare extracted columns against a list of original columns to ensure validity.

## E.2 Implementation Details

The algorithm is implemented using regular expressions to parse the SQL query. Key implementation details include:

- Use of `re.sub()` for comment removal and whitespace normalization.

- Application of `re.search()` and `re.findall()` for extracting column names from different parts of the query.

- Special treatment for columns used within functions in the `SELECT`, `WHERE`, `ORDER BY` clauses.

## E.3 An example of data-attribution tracking for Table QA

Here, we use the table transformation in Fig. 2–③ as an example to illustrate our data-attribution tracking algorithm (Fig. 11):

- **(a)** Adding the Tracking Index Column

- **(b)** Executing the SQL Command

- **(c)** Identifying Selected Rows

- **(d)** Parsing SQL Commands to Identify Selected Columns

- **(e)** Mapping to Original Indices

# F  Hallucinations in Sub-table Selection

Methods like CoTable and DATER aim to answer questions by performing complex table transformations—specifically, selecting sub-tables from the input table based on reasoning steps. However, these methods are prone to errors regarding which table entries to select, leading to irrational or irrelevant information being considered in the final answer.

As illustrated in Fig. 12, although Chain-of-Table correctly answers the question **Q**: *True or False? In four different baseball games, the final score was 9-2*, it irrationally selects unrelated information (game 3) from the input table. Similarly, DATER, shown in Fig. 13, selects rows 2, 3, 4, 5, and 9 to answer the same question. However, the inclusion of row 3 is illogical and does not contribute to a valid answer.

**Statement: the wildcats kept the opposing team scoreless in four games**

**Input Table: 1947 kentucky wildcats football team**

Step 1: f_select_row(row 1, row 2, row 3, row 4, row 8)

| game | date | opponent | result | wildcats points | opponents | record |
|---|---|---|---|---|---|---|
| 1 | sept 20 | ole miss | loss | 7 | 14 | 0 - 1 |
| 2 | sept 27 | cincinnati | win | 20 | 0 | 1 - 1 |
| 3 | oct 4 | xavier | win | 20 | 7 | 2 - 1 |
| 4 | oct 11 | 9 georgia | win | 26 | 0 | 3 - 1 , 20 |
| 5 | oct 18 | 10 vanderbilt | win | 14 | 0 | 4 - 1 , 14 |
| 6 | oct 25 | michigan state | win | 7 | 6 | 5 - 1 , 13 |
| 7 | nov 1 | 18 alabama | loss | 0 | 13 | 5 - 2 |
| 8 | nov 8 | west virginia | win | 15 | 6 | 6 - 2 |
| 9 | nov 15 | evansville | win | 36 | 0 | 7 - 2 |
| 10 | nov 22 | tennessee | loss | 6 | 13 | 7 - 3 |

Step 2: f_select_column(game, wildcats points, opponents)

| game | date | opponent | result | wildcats points | opponents | record |
|---|---|---|---|---|---|---|
| 2 | sept 27 | cincinnati | win | 20 | 0 | 1 - 1 |
| 3 | oct 4 | xavier | win | 20 | 7 | 2 - 1 |
| 4 | oct 11 | 9 georgia | win | 26 | 0 | 3 - 1 , 20 |
| 5 | oct 18 | 10 vanderbilt | win | 14 | 0 | 4 - 1 , 14 |
| 9 | nov 15 | evansville | win | 36 | 0 | 7 - 2 |

Step 3: f_sort_column(opponents)

| game | wildcats points | opponents |
|---|---|---|
| 2 | 20 | 0 |
| 3 | 20 | 7 |
| 4 | 26 | 0 |
| 5 | 14 | 0 |
| 9 | 36 | 0 |

Step 4: simple_query()

| game | wildcats points | opponents |
|---|---|---|
| 2 | 20 | 0 |
| 4 | 26 | 0 |
| 5 | 14 | 0 |
| 9 | 36 | 0 |
| 3 | 20 | 7 |

**Prompting LLM for the final answer... >>>**

**Prediction: TRUE**

Figure 12: Although CoTable correctly answers the question **Q**: *True or False? In four different baseball games, the final score was 9-2*, it irrationally selects unrelated information (game 3) from the input table.

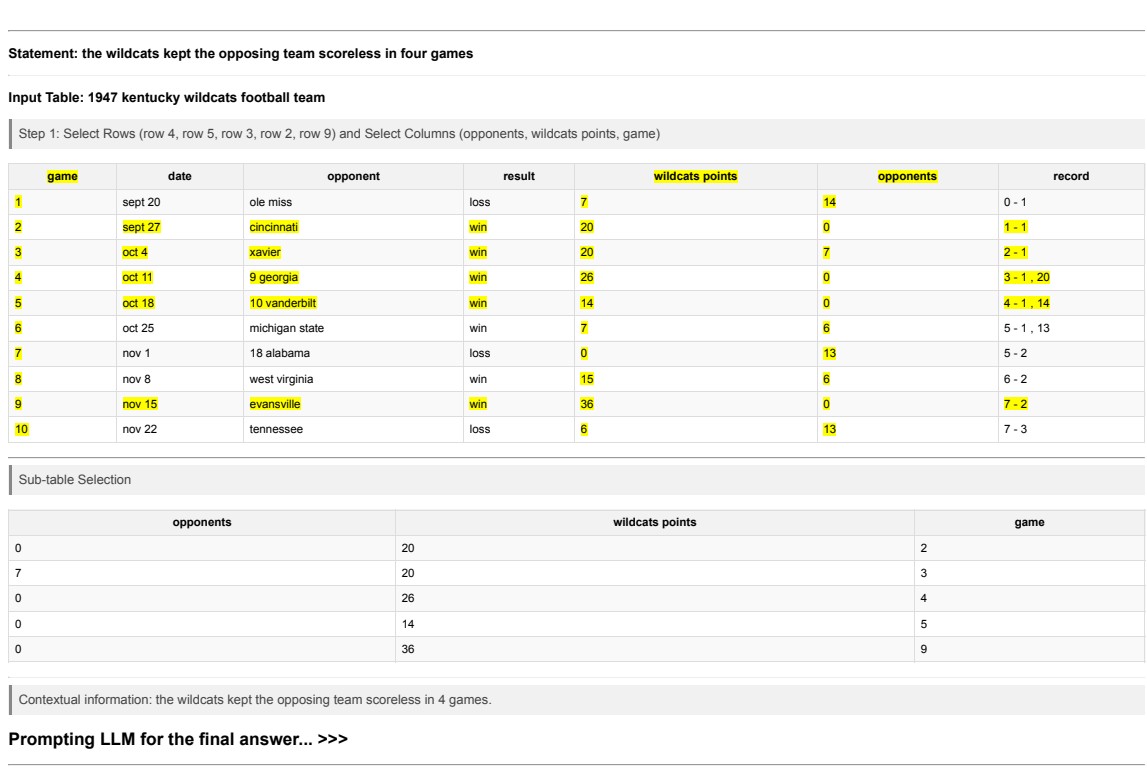

Figure 13: DATER selects rows 2, 3, 4, 5, and 9 to answer the question. However, the inclusion of row 3 is illogical and does not contribute to a valid answer.

# G   Details for LLM-as-a-Judge Experiments

In this section, we describe how we prompt LLMs to work as judges for evaluating explanation methods.

## G.1   Prompt to LLM Judges in Prediction Verification

---

**👤 Prompt for LLM-as-a-Judge in Prediction Verification**

---

prompt = f"""

The Table Fact Verification (TabFact) model is working on verifying if a given Statement is TRUE or FALSE based on a given input Table.

You are given an HTML file containing a Statement, Input Table, Prediction, and an Explanation clarifying the Prediction.

Your task is to carefully analyze the Explanation and determine whether the Prediction is correct or not.

Explanation Method: [**method**]

[**method_specific_info**]

HTML content: [**html_content**]

Answer with 'option1' or 'option2' only.

You MUST ignore the order of the options and answer based on the correctness of the Prediction!

"""

---

## G.2   Prompt to LLM Judges in Forward Simulation

---

**👤 Prompt for LLM-as-a-Judge in Forward Simulation**

---

prompt = f"""

Given an input statement, an Artificial Intelligence (AI) model will output either TRUE or FALSE. Your job in this Simulation task is to use the AI's explanation to guess the machine response. Specifically, please choose which response (TRUE/FALSE) model would output regardless of whether you think that response is correct or not.

Explanation: [**text_content**]

Based on this explanation, guess what the model will predict on the Statement based on the provided explanation. Answer with only 'TRUE' or 'FALSE':

"""

---

## G.3   Prompt to LLM Judges in Preference Ranking

It is well known that LLM-as-a-Judge exhibits a strong bias toward the position of the options presented to it (Dubois et al., 2024). To eliminate this bias in our prompt, we shuffle the order of the four methods four times and compute the average ranking.

---

👤 **Prompt for LLM-as-a-Judge in Preference Ranking**

---

```
    prompts = []

    num_methods = len(methods)
```

# Create a dictionary mapping methods to their descriptions

method_descriptions = {

"DATER": """DATER is a method that focuses on selecting relevant information from the input table and providing contextual information to support the statement verification process. The explanation contains:

1. Sub-table Selection: DATER selects a sub-table from the original input Table that is relevant to the Statement.

2. Contextual Information: DATER provides contextual information that is fact-checked against the Table.""",

"COT": """COT is a method that breaks down the question-answering process into a series of intermediate tables. Each step in the chain represents a specific operation on the table, leading to the final answer. The explanation contains:

1. Step Descriptions: Each step is accompanied by a function with arguments, providing context for the transformation.

2. Intermediate Tables: We display the intermediate tables resulting from each function, showing the state of the data at each step.

3. Row and Column Highlighting: Rows and Columns used in the current step are highlighted with background-color:yellow.""",

"Text2SQL": """Text2SQL is a method that translates the natural language question into a single SQL query. The SQL query itself serves as the explanation for how the system arrives at its answer. The explanation contains: The generated SQL command that will be directly applied onto the table to generate the final answer.""",

"POS": """POS is a Table QA method that breaks down the question-answering process into a series of natural-language steps. Each step represents a specific operation on the table, leading to the final answer. The explanation contains:

1. Step Descriptions: Each step is accompanied by a natural-language description of the atomic step performed, providing context for the transformation.

2. Intermediate Tables: We display the intermediate tables resulting from each step, showing the state of the data at each step.

3. Attribution Maps: We highlight the the rows, columns, and cells involved in each table transformation over intermediate tables. Row and Column Highlighting: Rows and Columns used in the current step are highlighted with background-color:yellow. Cell Highlighting: Cells that directly match the conditions in the current step are highlighted with background-color:90EE90.""" }

```
    for i in range(num_methods):
        shuffled_methods = methods[i:] + methods[:i]
```

prompt = f""" You are given explanations from four different methods for the same table fact verification task. Please rank these explanations based on their clarity, coherence, and helpfulness in understanding the model's reasoning.

Clarity Definition: How easy is the explanation to understand? Is the language clear and straightforward?

Coherence Definition: Does the explanation logically flow and make sense as a whole? Are the ideas well-connected?

Helpfulness in Understanding the Model's Reasoning Definition: How effectively does the explanation help you understand why the model made its decision? Does it reveal the reasoning process?

Provide the ranking from best to worst.

Explanations:
"""

---

# H Prompt Engineering

## H.1 Prompt for Atomic Planning

### H.1.1 Decomposition of Question $Q$

The decomposition process breaks down the complex question $Q$ into a sequence of atomic steps. This is achieved through a carefully crafted prompt provided to the LLM. The prompt includes:

- **Instructional Guidelines:** We instruct the LLM to "Develop a step-by-step plan to answer the question given the input table".

- **Emphasis on Atomicity:** The LLM is instructed that "Each step in your plan should be very atomic and straightforward, ensuring they can be easily executed or converted into SQL".

- **In-context Examples:** We provide example inputs $(T, Q)$ along with their corresponding plans to serve as in-context examples for planning (see Appendix I).

### H.1.2 Sequencing of Steps

Correct sequencing is crucial because each step depends on the output of the previous one. We ensure proper sequencing by:

- **Explicit Instructions:** The LLM is instructed that "The order of steps is crucial! You must ensure the orders support the correct information retrieval and verification!".

- **Dependencies:** Clarifying that "The next step will be executed on the output table of the previous step. The first step will be executed on the given Table".

- **Handling Comparatives and Superlatives:** Instructing the LLM on how to handle statements involving terms like 'highest', 'lowest', etc., by ordering the table before selecting rows.

---

**👤 Prompt for atomic planning**

[**In-context examples**]

**### Here come to your task!**

**Table caption:** {caption}

/* {table2string(table_info["table_text"])} */ # Convert Table into markdown format

**This Table has {num_rows} rows.**

**Statement:** {sample["statement"]}

Let's develop a step-by-step plan to verify if the given **Statement** is **TRUE** or **FALSE** on the given **Table**!

You **MUST** think carefully analyze the **Statement** and comprehend it before writing the plan!

**Plan Steps:** Each step in your plan should be very atomic and straightforward, ensuring they can be easily executed or converted into SQL.

You **MUST** make sure all conditions (except those mentioned in the table caption) are checked properly in the steps.

**Step order:** The order of steps is crucial! You must ensure the orders support the correct information retrieval and verification!

The next step will be executed on the output table of the previous step. The **first step** will be executed on the given **Table**.

For comparative or superlative **Statement** involving "highest," "lowest," "earliest," "latest," "better," "faster," "earlier," etc., you should order the table accordingly before selecting rows. This ensures that the desired comparative or superlative data is correctly retrieved.

**Plan:**

---

### H.1.3 The Importance of the Step Order

In this example, step 1 is crucial. If the table is not ordered by 'rank' first, selecting row number 1 (step 2) or filtering by 'athlete' (step 3) will return the wrong result. Only by ensuring that the table is correctly ordered beforehand can we reliably select the top-ranked athlete. Thus, the sequence of steps must be followed precisely to avoid logical errors.

---

**👤 A plan where the step order determines the correctness**

---

**Table:** Olympic 2018; Table Tennis

```
/*
col : rank| athlete              | time
row 1 : 1 | manjeet kaur (ind)    | 52.17
row 2 : 2 | olga tereshkova (kaz) | 51.86
row 3 : 3 | pinki pramanik (ind)  | 53.06
*/
```

**Statement:** manjeet had the highest rank in the competition.

**Plan:**

1. Order the table by 'rank' in ascending order.

2. Select row number 1.

3. Select rows where 'athlete' is 'manjeet' using the `LIKE` function.

4. Use a `CASE` statement to return TRUE if the number of rows is equal to 1, otherwise return FALSE.

---

## H.2 Prompt for Step-to-SQL

---

**👤 Prompt for Step-to-SQL**

---

[**In-context examples**]

Given this table:

`/* {table2string(intermediate_table)} */`

Data types of columns:

- {col_1}: {dtype_str_1}

- {col_2}: {dtype_str_2}

- …

Write an SQL command that: {natural_language_step}

The original table has {num_rows} rows.

**Constraints for your SQL:**

1. If using SELECT COUNT(*), SUM, MAX, AVG, you MUST use AS to name the new column. If adding new columns, they should be different than columns {existing_cols}.

2. Your SQL command **MUST** be compatible and executable by Python `sqlite3` and `pandas`.

3. If using FROM, the table to be selected **MUST** be {table_name}.

---

# I In-context Examples

## I.1 In-context Examples for Atomic Planning

> ** 👤 In-context examples for atomic planning**
>
> **TabFact**
>
> **Table:** 2005 tournament results
>
> ```
> /*
> col  : id | name    | hometown    | score
> row 1 : 1 | alice   | new york    | 85
> row 2 : 2 | bob     | los angeles | 90
> row 3 : 3 | charlie | chicago     | 75
> row 4 : 4 | dave    | new york    | 88
> row 5 : 5 | eve     | los angeles | 92
> */
> ```
>
> **Statement:** in 2005 tournament, bob and charlie are both from chicago.
>
> **Plan:** # Natural-language step
>
> 1. Select rows where the 'name' is 'bob' or 'charlie'.
>
> 2. Select rows where 'hometown' is 'chicago'.
>
> 3. Use a `CASE` statement to return TRUE if the number of rows is equal to 2, otherwise return FALSE.
>
> **WikiTQ**
>
> **Table:** 2005 tournament results
>
> ```
> /*
> col  : id | name    | hometown    | score
> row 1 : 1 | alice   | new york    | 85
> row 2 : 2 | bob     | los angeles | 90
> row 3 : 3 | charlie | chicago     | 75
> */
> ```
>
> **Question:** which players are from chicago?
>
> **Plan:** # Natural-language step
>
> 1. Select rows where the 'hometown' is 'chicago'.
>
> 2. Select the 'name' column.

### I.2 In-context Examples for Step-to-SQL

---

👤 **In-context examples for Step-to-SQL**

---

Given this table:

```
/*
col : id  | name    | department | salary | years
row 1 : 1 | alice   | it         | 95000  | 3
row 2 : 2 | bob     | finance    | 105000 | 5
row 3 : 3 | charlie | marketing  | 88000  | 2
*/
```

Write an SQL command that: Select rows where the 'salary' is greater than 100000.

SQL is:

**SELECT** *
**FROM** table_sql
**WHERE** salary > 100000;
*-- Select rows where the 'salary' is greater than 100000.*

---

## J   Error Analysis of `POS` and Improved Planning Algorithm

### J.1   Error Analysis of `POS`

We notice that many errors in `POS` are due to the planning stage rather than the Step-to-SQL process. In particular, the Planner misses condition checks (see Fig. 14, Fig. 16, Fig. 15, Fig. 17, Fig. 18) in atomic steps. Another interesting (and inherently unavoidable) error is presented in Fig. 19. Arguably, this error can be attributed to the planner, as it has access to both the query and the input table and should therefore generate the correct step. In this case, the Step-to-SQL component is functioning as intended. We also offer an alternative perspective on this interesting failure in Appendix J.2.

### J.2   Quantifying contributions of Planning and Step-to-SQL to `POS` errors

In the example shown in Fig. 19, the failure to retrieve the correct rows is not due to any flaw in our Step-to-SQL converter. Instead, it stems from a mismatch between the query token and the table's content—and, ultimately, from an inconsistent ground-truth label. Specifically, the NL Planner faithfully mirrors the input string "bjørn" when it generates its first step ("Select rows where player = 'thomas bjørn'"), but the actual table only contains "thomas bjarn," so no rows are returned. Under standard Table-QA semantics, "bjørn" and "bjarn" denote two distinct entities, so the correct label for this query should be **False**, not **True**. Consequently, neither the Step-to-SQL module nor the NL Planner has malfunctioned; the discrepancy arises from the dataset's labeling and the lack of accent normalization in the query definition.

Putting this example in context, we observe that across all of our error analyses there are **no instances** where the Step-to-SQL module itself directly produces an incorrect SQL query. Instead, errors occur at two other points in the pipeline:

- NL Planner errors: the natural-language plan deviates from the correct reasoning (even though every resulting SQL statement remains executable), or

- Fallback errors: at least one SQL in the plan is invalid, causing the system to defer to the end-to-end QA model, which then also fails to produce the right answer.

To quantify how much each of these failure modes contributes to our overall error rate on TabFact and WikiTQ, we labeled every incorrect sample in our evaluation set as either a Planning error or a Fallback error. The breakdown of their relative frequencies is reported in Tab. 12.

Table 12: Contribution to `POS` errors of Planning and Fallback.

| Dataset | Model | Planning (%) | Fallback (%) |
|---------|-------|-------------|-------------|
| TabFact | `GPT-4o-mini` | 95.4 | 4.6 |
| TabFact | `GPT-3.5` | 89.7 | 10.3 |
| WikiTQ | `GPT-4o-mini` | 75.5 | 24.5 |
| WikiTQ | `GPT-3.5` | 70.7 | 29.3 |

We observe that planning mistakes constitute the vast majority of errors: over 95% for `GPT-4o-mini` on TabFact (and nearly 90% for GPT-3.5), and around 70–75% on the more complex WikiTQ benchmark. This confirms that *planning* is the primary bottleneck in `POS`.

### J.3   Planning one step at at time improves Table QA accuracy

Based on this observation, we implement an improved planning algorithm in which *only one step is generated at a time*, rather than generating all steps upfront, as shown in Fig. 1. This approach encourages the LLM to think one step at a time and reduces the complexity of the planning task. The input of NL Planner is the previous steps and the current intermediate table.

As shown in Tab. 13, planning one step at a time leads to substantial improvements in accuracy for both TabFact and WikiTQ with `GPT-4o-mini`. There are leading hypotheses for these improvements:

- **Grounding in real data.** Before generating each new step, one-step planning provides the actual intermediate table (the result of the previous SQL) into the prompt. This lets the LLM see exactly which rows and values it's operating on (condition on the actual intermediate table state), mitigating hallucinations and mis-alignments. By contrast, generating an entire multi-step plan up front requires the model to imagine how each intermediate table will look—and if those imagined tables don't match reality, later steps can go wrong.
- **Focused context window.** By feeding the LLM only the rows and columns it needs for each step—rather than the entire table—we keep the input prompt concise and free of irrelevant data. Since LLMs can struggle when their context grows too large Li et al. (2024), this focused context helps them generate the next step more accurately.

Table 13: Planning one step at a time (one-step planning) with `GPT-4o-mini`.

| Method | TabFact (%) | WikiTQ (%) |
|---|---|---|
| End-to-end QA | 71.17 | 49.24 |
| POS one-**time** planning | 77.22 | 48.90 |
| POS one-**step** planning | 83.45 | 59.32 |

Figure 14: `POS` predicts TRUE but the groundtruth is FALSE (False Positive). In planning, `POS` misses checking the player name.

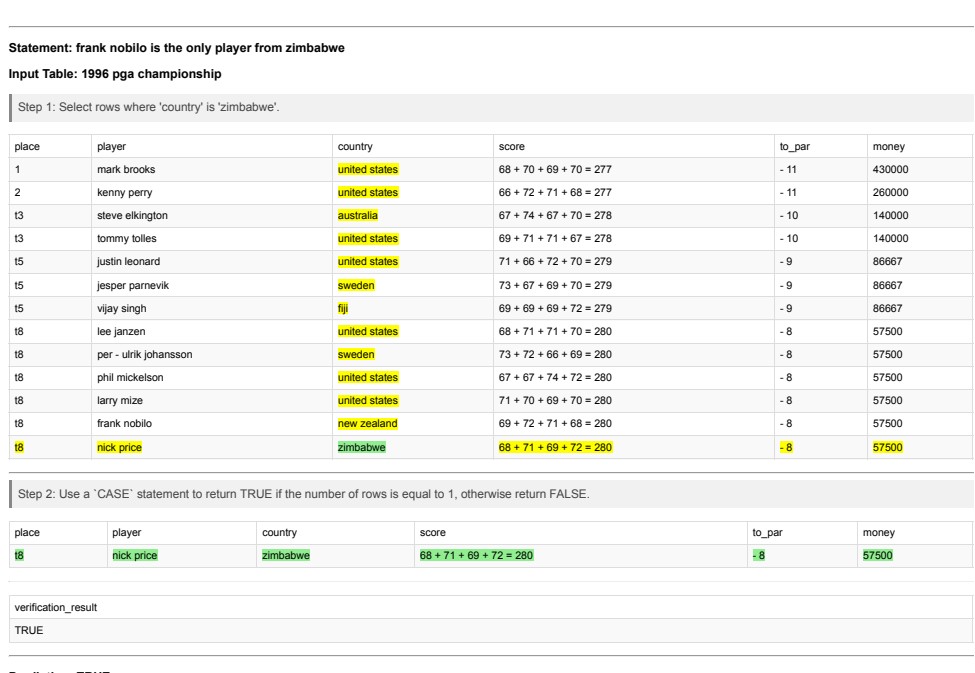

Figure 15: **POS** predicts TRUE but the groundtruth is FALSE (False Positive). In planning, **POS** misses checking the player name.

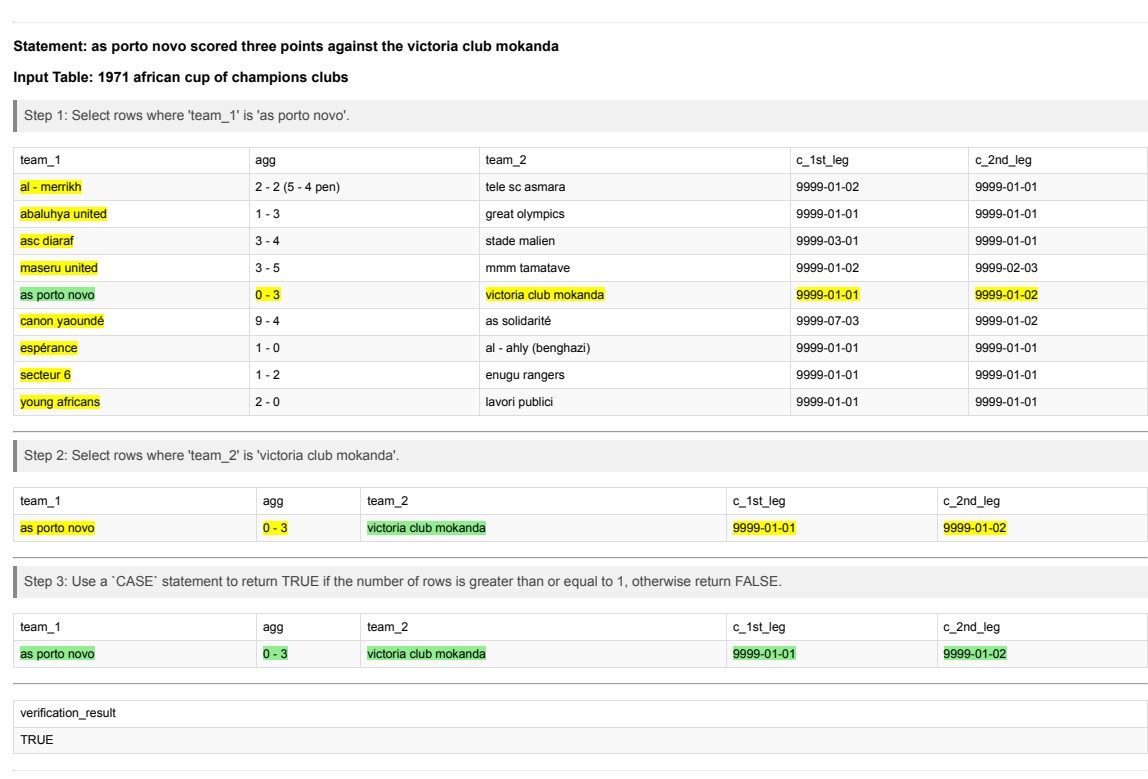

Figure 16: `POS` predicts TRUE but the groundtruth is FALSE (False Positive). In planning, `POS` misses checking the score.

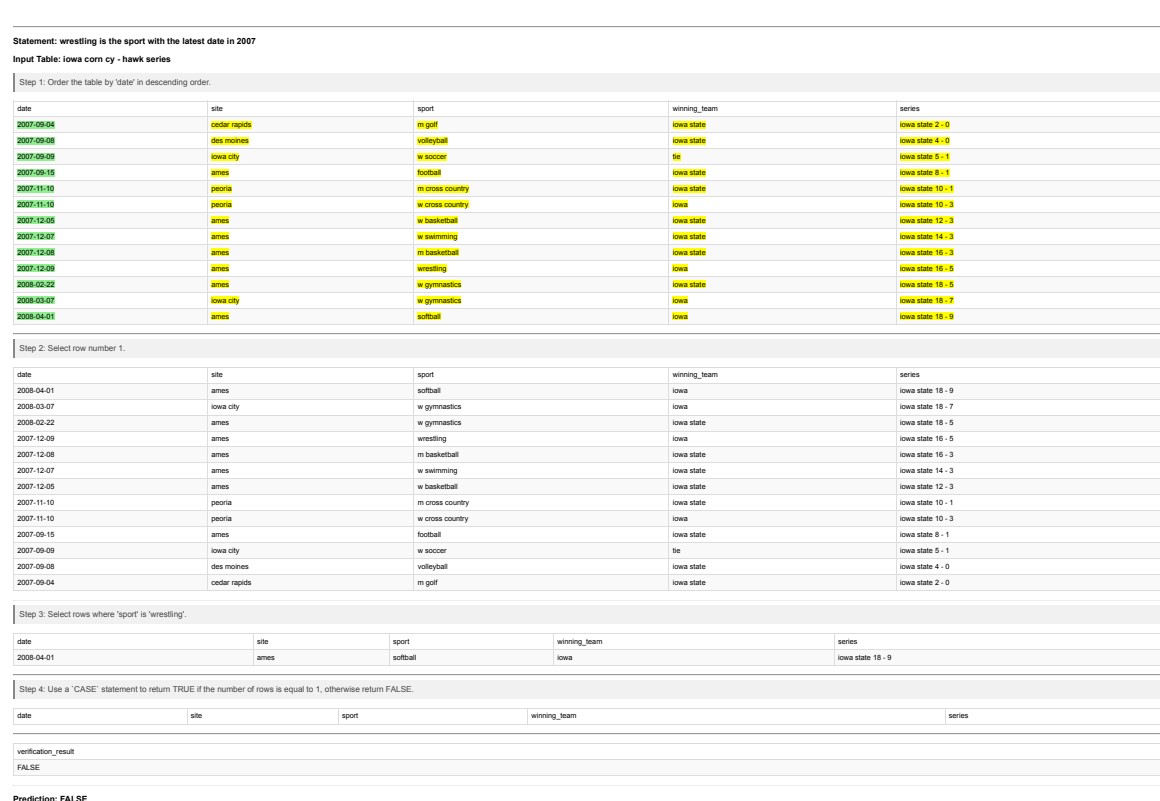

Figure 17: **POS** predicts FALSE but the groundtruth is TRUE (False Negative). In planning, **POS** misses checking the year.

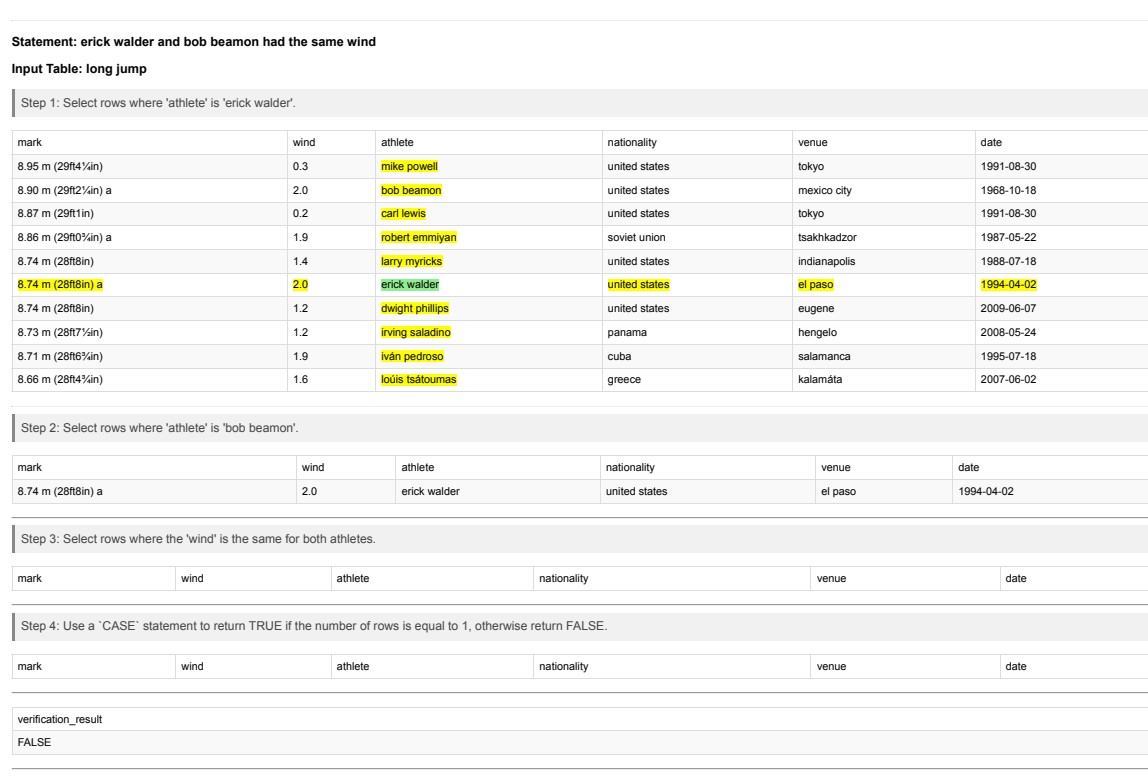

Figure 18: **POS** predicts FALSE but the groundtruth is TRUE (False Negative). In planning, **POS** should select two rows at the same step.

**Statement: thomas bjørn , of denmark , has a score of 68 + 71 + 76 = 215**

**Input Table: 1998 open championship**

Step 1: Select rows where 'player' is 'thomas bjørn'.

| place | player | country | score | to_par |
|---|---|---|---|---|
| 1 | brian watts | united states | 68 + 69 + 73 = 210 | e |
| t2 | jim furyk | united states | 70 + 70 + 72 = 212 | + 2 |
| t2 | mark o'meara | united states | 72 + 68 + 72 = 212 | + 2 |
| t2 | jesper parnevik | sweden | 68 + 72 + 72 = 212 | + 2 |
| 5 | justin rose (a) | england | 72 + 66 + 75 = 213 | + 3 |
| t6 | thomas bjärn | denmark | 68 + 71 + 76 = 215 | + 5 |
| t6 | brad faxon | united states | 67 + 74 + 74 = 215 | + 5 |
| t6 | john huston | united states | 65 + 77 + 73 = 215 | + 5 |
| t6 | tiger woods | united states | 65 + 73 + 77 = 215 | + 5 |
| t10 | david duval | united states | 70 + 71 + 75 = 216 | + 6 |
| t10 | costantino rocca | italy | 72 + 74 + 70 = 216 | + 6 |
| t10 | raymond russell | scotland | 68 + 73 + 75 = 216 | + 6 |
| t10 | katsuyoshi tomori | japan | 75 + 71 + 70 = 216 | + 6 |

Step 2: Select rows where 'country' is 'denmark'.

| place | player | country | score | to_par |
|---|---|---|---|---|

Step 3: Select rows where 'score' is '68 + 71 + 76 = 215'.

| place | player | country | score | to_par |
|---|---|---|---|---|

Step 4: Use a `CASE` statement to return TRUE if the number of rows is equal to 1, otherwise return FALSE.

| place | player | country | score | to_par |
|---|---|---|---|---|

| verification_result |
|---|
| FALSE |

**Prediction: FALSE**

Figure 19: `POS` predicts FALSE but the groundtruth is TRUE (False Negative). Arguably, this error can be attributed to the planner, as it sees both the query and the input table and **should** therefore generate the correct step (i.e., Select rows where 'player' is 'thomas bjarn'). We observe that the Step-to-SQL component always functions correctly. We also offer an alternative perspective on this interesting failure in Appendix J.2.

