# OpenReview forum: "Interpretable LLM-based Table Question Answering"
_TMLR — Accepted by TMLR_

### Review · Reviewer_Rtym · 2025-05-02

**Summary Of Contributions:**

This paper introduces Plan-of-SQLs (POS), a novel LLM-based Table Question Answering (Table QA) method focused on interpretability. The core idea is to decompose a natural language question into a sequence of atomic, natural language steps. Each step is then translated into a simple, executable SQL query that transforms the table incrementally. This process ensures transparency as each intermediate table state and the corresponding SQL operation are explicitly shown.
The contributions of the paper are:
Introduction of POS: A new interpretable Table QA method that breaks down complex questions into atomic, understandable steps translated into SQL queries. This contrasts with methods that generate complex single SQL queries or use opaque LLM reasoning steps (like abstract function calls).
Enhanced Interpretability: Through experiments involving both human participants and LLM-based judges, the authors demonstrate that POS explanations are superior in quality (preference ranking) and significantly improve users' ability to simulate the model's behavior (forward simulation) and verify its predictions compared to existing interpretable methods (Text-to-SQL, DATER, CoTable, self-explanation).
Competitive Performance and Efficiency: POS achieves QA accuracy comparable to state-of-the-art methods on standard benchmarks (TabFact, WikiTQ, FeTaQA) while being significantly more efficient, requiring substantially fewer LLM calls (up to 25x fewer than DATER) and database queries (up to 25x fewer than Binder). It also shows better robustness on large tables where other methods often struggle.
LLM-as-Human-Proxy Validation: The study reveals high agreement (up to 90.59% in forward simulation) between LLM judges and human evaluators when assessing POS explanations, suggesting that LLMs can serve as reliable proxies for human judgment in evaluating Table QA explanation quality, potentially reducing the cost of user studies.
Generation of Attribution Maps: POS inherently generates step-by-step attribution maps by highlighting the rows/columns used and cells satisfying conditions in each SQL step, providing a clear trace of the reasoning process.
Ablation Study: An ablation study confirms the importance of SQL execution and natural language planning steps for the interpretability benefits of POS.

**Audience:**

Yes

**Broader Impact Concerns:**

The paper focuses on improving the interpretability of Table Question Answering systems, which is generally seen as a positive step towards more responsible and trustworthy AI. By making the reasoning process transparent through atomic SQL steps, the work directly aims to mitigate risks associated with opaque AI decision-making, particularly in high-stakes domains like finance and healthcare, which the authors explicitly mention.
Therefore, the work itself does not appear to introduce new ethical concerns or negative societal impacts

**Claims And Evidence:**

Yes

**Requested Changes:**

Expand on Planner Error Analysis (Appendix J / Sec 5): While Appendix J shows examples of planning errors (missed conditions), it would be beneficial to provide a more quantitative analysis if possible. For instance, what percentage of errors on TabFact/WikiTQ are attributable to planning vs. Step-to-SQL errors? This would give a clearer picture of the main bottleneck. Discussing why the one-step planning approach (Table 11) yields such significant accuracy improvements (+6% on TabFact, +10% on WikiTQ) beyond just "reducing complexity" would also be valuable. Does it allow the LLM to condition on the actual intermediate table state more effectively?

Discuss Handling More Complex SQL Needs: Briefly discuss the limitations of the current atomic SQL definition. Are there common Table QA reasoning patterns that POS cannot currently handle due to the restriction on SQL complexity (e.g., queries requiring explicit JOINs, complex aggregations with GROUP BY and HAVING, window functions)? Acknowledging these potential limitations and outlining how POS might be extended (even if as future work) would strengthen the discussion.

Clarify Fallback Mechanism Usage: Section 5 mentions a fallback mechanism (end-to-end LLM) triggered in 3.16% (TabFact) and 13.58% (WikiTQ) of cases due to SQL execution failures. Could the authors clarify if the main accuracy results reported in Table 6 include results from this fallback? If so, it slightly nuances the "program-only" nature for those specific samples. It's a minor point but worth clarifying for full transparency. If the fallback is used, how does its accuracy compare to the main POS approach?

**Strengths And Weaknesses:**

Strengths:
Well-Designed Method (POS): The POS method is intuitive and well-motivated. Enforcing atomicity (single condition, single variable per SQL step) and using natural language for planning before SQL conversion makes the process inherently more understandable and less prone to complex errors compared to end-to-end Text-to-SQL or opaque LLM reasoning steps.

Efficiency and Robustness: POS demonstrates significant improvements in computational efficiency (fewer LLM calls and DB queries) and robustness to large table sizes compared to several state-of-the-art methods. This makes it more practical for real-world applications.


Weaknesses:
Dependence on LLM Planner Quality: The paper acknowledges (Sec 5, Conclusion, Appendix J) that POS's accuracy is limited by the LLM's ability to generate a correct plan of atomic steps. Errors in the initial natural language plan (e.g., missing conditions, incorrect logic) directly lead to incorrect final answers, even if the subsequent Step-to-SQL translation is perfect. While the one-step-at-a-time planning shows improvement (Appendix J), the fundamental reliance on the LLM's planning capability remains a potential bottleneck.

Handling "Messy" Real-World Tables: The evaluation is primarily on benchmark datasets that are relatively clean. The paper briefly discusses (Sec 5) extending POS to handle "messy" tables (merged cells, irregular formats) using preprocessing tools like NormTab, but this is presented as future work and not empirically validated. The practical applicability to truly unstructured or noisy real-world tables is not fully demonstrated.

Limited Scope of "Atomicity": While the defined "atomic" steps (single clause, max one condition/variable) enhance interpretability, they might be overly restrictive for certain complex reasoning patterns that could potentially be expressed clearly in slightly more complex SQL (e.g., joins, nested queries, more complex WHERE clauses). The trade-off between strict atomicity and expressive power could be explored further.

SQL Generation Errors: Although POS aims to simplify SQL generation, errors can still occur, especially with exact matching issues (e.g., 'bjørn' vs 'bjon', Fig 19). The paper mentions a fallback mechanism but doesn't deeply analyze the frequency or types of Step-to-SQL errors.

Interpretability in Free-form QA: The extension to FeTaQA involves a final LLM call to generate the free-form answer, which re-introduces a black-box component, slightly compromising the end-to-end interpretability claim for this specific task type. The paper notes this limitation (Sec C).

---

> ### Author Response · Authors · 2025-05-06
> **We sincerely appreciate your thoughtful feedback and here is our responses!**
>
> We thank the Reviewer `Rtym` for the valuable and insightful feedback!
>
> Below, please see our responses to your requested changes.

---

> > ### Author Response · Authors · 2025-05-06
> > **Authors' Response to Requested Change 1**
> >
> > > **Requested Change 1**:
> > Dependence on LLM Planner Quality: The paper acknowledges (Sec 5, Conclusion, Appendix J) that POS's accuracy is limited by the LLM's ability to generate a correct plan of atomic steps. Errors in the initial natural language plan (e.g., missing conditions, incorrect logic) directly lead to incorrect final answers, even if the subsequent Step-to-SQL translation is perfect. While the one-step-at-a-time planning shows improvement (Appendix J), the fundamental reliance on the LLM's planning capability remains a potential bottleneck.
> >
> > > Expand on Planner Error Analysis (Appendix J / Sec 5): While Appendix J shows examples of planning errors (missed conditions), it would be beneficial to provide a more quantitative analysis if possible.
> >
> > > Q.1.1: For instance, what percentage of errors on TabFact/WikiTQ are attributable to planning vs. Step-to-SQL errors? This would give a clearer picture of the main bottleneck
> >
> > Thank you for this very thoughtful comment. To quantify the relative impact of planning versus Step-to-SQL on POS error, we performed an error analysis of incorrect samples from TabFact and WikiTQ under both GPT3.5 and GPT4o-mini, labeling each sample as either a planning error or a Step-to-SQL error.
> > - **Planning error**: Every Step-to-SQL translation along the plan was correct and executable, but the natural-language plan was wrong.
> > - **Step-to-SQL error**: At least one SQL in the plan of SQLs is erroneous.
> >
> > We show the results of the analysis in the following table.
> >
> > **Table**: The contributions (%) to POS errors of Planning and Step-to-SQL module.
> >
> > | **Dataset** | **Model**    | **Planning (%)** | **Step-to-SQL (%)** |
> > |-------------|--------------|------------------|---------------------|
> > | TabFact     | GPT4o-mini   | 95.4%            | 4.6%                |
> > | TabFact     | GPT3.5       | 89.7%            | 10.3%               |
> > | WikiTQ      | GPT4o-mini   | 75.5%            | 24.5%               |
> > | WikiTQ      | GPT3.5       | 70.7%            | 29.3%               |
> >
> > **We found that planning is indeed the primary bottleneck**.
> > - On TabFact, over 95% of GPT4o-mini’s errors (and nearly 90% for GPT3.5) trace back to flawed plans.
> > - Even on the more complex WikiTQ benchmark, planning still accounts for ~70–75% of errors.
> > - “One-step-at-a-time” planning (`Appendix J`) yields substantial accuracy improvements on both benchmarks, demonstrating that more robust planning will significantly drive the improvements in POS performance while preserving our interpretability advantages.
> >
> > We have added this analysis into `Appendix J.2` of our revised manuscript (new text in teal color).
> >
> > > Q1.2. Discussing why the one-step planning approach (Table 11) yields such significant accuracy improvements (+6% on TabFact, +10% on WikiTQ) beyond just "reducing complexity" would also be valuable. Does it allow the LLM to condition on the actual intermediate table state more effectively?
> >
> > Thank you for this great suggestion!
> >
> > We have expanded our discussion for one-step planning in `Appendix J.3` to make it more clear why one-step planning is more effective. Beyond “reducing planning complexity”, we list down two leading hypotheses:
> >
> > 1. **Grounding in real data**. Before generating each new step, one-step planning provides the actual intermediate table (the result of the previous SQL) into the prompt. This lets the LLM see exactly which rows and values it’s operating on (condition on the actual intermediate table state), mitigating hallucinations and mis‐alignments. By contrast, generating an entire multi-step plan up front requires the model to imagine how each intermediate table will look—and if those imagined tables don’t match reality, later steps can go wrong.
> > 2. **Focused context window**. By feeding the LLM only the rows and columns it needs for each step—rather than the entire table—we keep the input prompt concise and free of irrelevant data. Since LLMs can struggle when their context grows too large [1], this focused context helps them generate the next step more accurately.
> >
> >
> > [1] Long-context LLMs Struggle with Long In-context Learning, Li et al, 2024.

---

> > > ### Author Response · Authors · 2025-05-06
> > > **Authors' Response to Requested Change 2**
> > >
> > > > **Requested Change 2**:
> > > Limited Scope of "Atomicity": While the defined "atomic" steps (single clause, max one condition/variable) enhance interpretability, they might be overly restrictive for certain complex reasoning patterns that could potentially be expressed clearly in slightly more complex SQL (e.g., joins, nested queries, more complex WHERE clauses). The trade-off between strict atomicity and expressive power could be explored further.
> > > Discuss Handling More Complex SQL Needs: Briefly discuss the limitations of the current atomic SQL definition. Are there common Table QA reasoning patterns that POS cannot currently handle due to the restriction on SQL complexity (e.g., queries requiring explicit JOINs, complex aggregations with GROUP BY and HAVING, window functions)? Acknowledging these potential limitations and outlining how POS might be extended (even if as future work) would strengthen the discussion.
> > >
> > >
> > > Thank you for this insightful question!
> > >
> > > In POS, every table question is decomposed into a linear sequence of atomic steps, each implementing exactly one of the following SQL built-in primitives:
> > >
> > > ```
> > > - Filter (WHERE)
> > > - Select (pick columns via SELECT)
> > > - Group & aggregate (GROUP BY, COUNT, SUM, etc.)
> > > - Sort & limit (ORDER BY, LIMIT)
> > > - Join (JOIN)
> > > - Set operations (UNION, INTERSECT, EXCEPT), etc.
> > > ```
> > > These are SQL’s basic operations—and because SQL is relationally complete (i.e., it can express every relational-algebra query) [2], one can build any SQL program, from simple counts to complex multi-table analytics, by simply chaining atomic SQL operations.
> > >
> > > [2] https://www.cwblogs.com/posts/relational-algebra/, last access 05/03/2025.
> > >
> > > Here is an example:
> > >
> > > | employee_id | name  | age | status   | department  | salary  |
> > > |-------------|-------|-----|----------|-------------|---------|
> > > | 1           | Alice | 45  | active   | Engineering | 120000  |
> > > | 2           | Bob   | 38  | active   | Sales       | 90000   |
> > > | 3           | Carol | 52  | inactive | HR          | 80000   |
> > > | 4           | David | 47  | active   | Engineering | 110000  |
> > > | 5           | Eve   | 41  | active   | Marketing   | 95000   |
> > > | 6           | Frank | 50  | active   | Sales       | 105000  |
> > > | 7           | Grace | 44  | active   | HR          | 98000   |
> > > | 8           | Heidi | 39  | active   | Marketing   | 87000   |
> > > | 9           | Ivan  | 55  | inactive | Engineering | 130000  |
> > > | 10          | Judy  | 60  | active   | Sales       | 115000  |
> > >
> > > For a complex query like: “Find the top 3 departments by average salary among active employees over 40”
> > >
> > > POS decomposes the query into:
> > >
> > > ```markdown
> > > 1. Filter rows where age > 40
> > > 2. Filter rows where status = 'active'
> > > 3. Group by department
> > > 4. Compute AVG(salary)
> > > 5. Sort by that average
> > > 6. LIMIT 3
> > > ```
> > >
> > > Each step becomes one standalone SQL query whose result feeds the next step.
> > >
> > > However, we admit that **there’s one class of questions that SQLs can’t handle end-to-end**: free-form queries that require generating a narrative or summary. For example:
> > > “Summarize the basic information of the football clubs in Saint Petersburg.”
> > >
> > > For these, we use atomic SQL operations to transform the input table, then offload only the final writing step to an LLM to turn those results into free-form answers.
> > >
> > > We have added a discussion of atomic-step coverage: “Can every table-based question be decomposed into a set of atomic steps?” in `Appendix C` of our revised manuscript.

---

> > > > ### Author Response · Authors · 2025-05-06
> > > > **Authors' Response to Requested Change 3**
> > > >
> > > > > **Requested Change 3**:
> > > > Clarify Fallback Mechanism Usage: Section 5 mentions a fallback mechanism (end-to-end LLM) triggered in 3.16% (TabFact) and 13.58% (WikiTQ) of cases due to SQL execution failures.
> > > > Could the authors clarify if the main accuracy results reported in Table 6 include results from this fallback? If so, it slightly nuances the "program-only" nature for those specific samples. It's a minor point but worth clarifying for full transparency. If the fallback is used, how does its accuracy compare to the main POS approach?
> > > > Thank you for this great comment. You’re right that our “program-only” pipeline does in fact rely on an end-to-end LLM when SQL execution fails, so it’s worth making this explicit:
> > > >
> > > > Thank you for this great comment! You’re right that our “program-only” pipeline does in fact rely on an end-to-end LLM when SQL execution fails, so it’s worth making this explicit:
> > > >
> > > > - **Inclusion in Table 6.**
> > > > All numbers reported for POS in Table 6 are overall accuracies after applying the fallback. Concretely, for each dataset we compute:
> > > >
> > > > $$
> > > > \text{Overall Acc} = (1 - r)\times A_{\text{POS}} \ + \ r\times A_{\text{LLM}}
> > > > $$
> > > >
> > > > where $r$ is the fallback rate (3.16% on TabFact, 13.58% on WikiTQ), $A_{\text{POS}}$ is the accuracy of the pure‐SQL pipeline, and $A_{\text{LLM}}$ is the end‐to‐end LLM accuracy.
> > > >
> > > > - **Effect on “program-only” claim.**
> > > > We follow the same strategy as DATER, Binder, and CoTable in backing off to an end-to-end model when the programmatic execution fails. We would like to clarify and agree with the Reviewer that the "program-only" nature for those specific samples is nuanced.
> > > >
> > > > In `Section 5` of our revised manuscript, we have clarified that Table 6 uses the weighted-sum formula above and fallback nuances the "program-only" nature for some samples.
> > > >
> > > > - **Overall vs. POS (program-only) accuracy.**
> > > > Table 6 shows that POS outperforms the end‐to‐end LLM significantly. Because the fallback approach has lower accuracy, blending POS vs. fallback will lower the combined accuracy compared to the POS‐only accuracy on the program‐handled samples.
> > > >
> > > > If you have any additional questions or concerns, please let us know. We’re more than happy to engage and clarify any points of our paper!

---

### Review · Reviewer_8AWs · 2025-05-02

**Summary Of Contributions:**

This paper introduces Plan-of-SQLs (POS) for Table Question Answering. POS uses LLM to decompose a Table QA problem into atomic steps which are then converted to SQL commands. The SQL commands are executed to retrieve the answer. POS also derives a chain of attribution maps as explanation for the produced solution. Experiments show that POS improves solution interpretability while maintaining accuracy compared to existing approaches. POS further reduces the number of required LLM calls, reducing cost.

**Audience:**

Yes

**Claims And Evidence:**

Yes

**Requested Changes:**

Can the authors comment on
- Why a DSL like SQL is chosen instead of a general language like python.
- Limitation (or the lack thereof) on the questions POS can answer due to its reliance on SQL.

**Strengths And Weaknesses:**

**Strength**

The paper is well-written and easy to follow.

The paper conducts thorough experiments against an extensive set of baselines including Dater and chain-of-table.

POS enhances solution interpretability, significantly reduces the number of required LLM calls while maintaining competitive accuracy.

**Weaknesses**

Conceptually the framework lacks novelty, for example, PaL [1] breaks a question down into steps which are represented by python operations, the final solution is then obtained by executing the python program.

The type of questions POS can answer may be limited by the SQL commands, as the answer must be directly derived from SQL execution. Can every Table-based question be decomposed into a set of atomic steps?

[1] Gao, Luyu, et al. "Pal: Program-aided language models." International Conference on Machine Learning. PMLR, 2023.

---

> ### Author Response · Authors · 2025-05-06
> **Thank you and here is our rebuttal!**
>
> We sincerely thank the Reviewer `8AWs` for the feedback and insightful comments!
>
> Please find our responses to each of your requested changes below.

---

> > ### Author Response · Authors · 2025-05-06
> > **Authors' Response to the connection between PAL and POS**
> >
> > > Weakness: Conceptually the framework lacks novelty, for example, PaL [1] breaks a question down into steps which are represented by python operations, the final solution is then obtained by executing the python program.
> > [1] Gao, Luyu, et al. "Pal: Program-aided language models." International Conference on Machine Learning. PMLR, 2023.
> >
> > Thank you for pointing us to Program-Aided Language models (PAL) [1]. While POS shares PAL’s idea of breaking a question into executable steps, it differs in ways that make it conceptually different and practically tailored to Table QA:
> > 1.	Atomicity
> > - PAL targets general arithmetic and symbolic reasoning via generating a single Python program (code + comments) to solve each problem .
> > - POS, by contrast, is specifically designed for Table QA. We decompose each question into atomic natural-language steps (we emphasize on the atomicity to help human users understand steps easier). Each step with at most one condition and one variable—ensuring each transformation is minimal, reliable, and easily understood by users.
> >
> > Atomicity is especially important with large tables. Imagine trying to filter rows where age > 30 and status = “active”—a user would have to juggle two conditions at once over dozens to thousands of entries, and it’s easy to get lost. By contrast, POS would do:
> > - Step 1: “Select rows where age > 30.”
> > - Step 2: “Select rows where status = ‘active’.”
> >
> > Each step translates into its own small SQL query and runs in order, so one can read and verify the result before moving on.
> > In PAL, atomicity would not matter much—adding more conditions/clauses ends up with a straightforward calculation. Here is an example from PAL: Roger started with 5 tennis balls. He later had 2 cans of 3 tennis balls. 5 + (2x3) = 11. The total is 11.
> >
> > [1] Gao, Luyu, et al. "Pal: Program-aided language models." International Conference on Machine Learning. PMLR, 2023.
> >
> > 2.	Faithful and granular explanations
> >
> > - PAL produces code whose internal logic may be not easy for end users to inspect or simulate without reading Python (e.g. lay users who do not have python expertise).
> > - POS generates a chain of attribution maps over intermediate tables (our Sec. 3.3), highlighting exactly which rows/columns/cells each step uses.
> >
> > In short, the two methods differ in how they explain their reasoning to users.
> >
> > Based on this discussion, we’ve added a paragraph on PAL in `Appendix B` of the revised manuscript (added text is in teal color).

---

> > > ### Author Response · Authors · 2025-05-06
> > > **Authors' Response to Requested Change 1**
> > >
> > > > **Requested Change 1**:The type of questions POS can answer may be limited by the SQL commands, as the answer must be directly derived from SQL execution. Can every Table-based question be decomposed into a set of atomic steps?
> > >
> > >
> > > Thank you for this insightful question. In POS, every table question is decomposed into a linear sequence of atomic steps, each implementing exactly one of the following SQL built-in primitives:
> > >
> > > ```
> > > - Filter (WHERE)
> > > - Select (pick columns via SELECT)
> > > - Group & aggregate (GROUP BY, COUNT, SUM, etc.)
> > > - Sort & limit (ORDER BY, LIMIT)
> > > - Join (JOIN)
> > > - Set operations (UNION, INTERSECT, EXCEPT), etc.
> > > ```
> > > These are SQL’s basic operations—and because SQL is relationally complete (i.e., it can express every relational-algebra query) [2], one can build any SQL program, from simple counts to complex multi-table analytics, by simply chaining atomic SQL operations.
> > >
> > > [2] https://www.cwblogs.com/posts/relational-algebra/, last access 05/03/2025.
> > >
> > > Here is an example:
> > >
> > > | employee_id | name  | age | status   | department  | salary  |
> > > |-------------|-------|-----|----------|-------------|---------|
> > > | 1           | Alice | 45  | active   | Engineering | 120000  |
> > > | 2           | Bob   | 38  | active   | Sales       | 90000   |
> > > | 3           | Carol | 52  | inactive | HR          | 80000   |
> > > | 4           | David | 47  | active   | Engineering | 110000  |
> > > | 5           | Eve   | 41  | active   | Marketing   | 95000   |
> > > | 6           | Frank | 50  | active   | Sales       | 105000  |
> > > | 7           | Grace | 44  | active   | HR          | 98000   |
> > > | 8           | Heidi | 39  | active   | Marketing   | 87000   |
> > > | 9           | Ivan  | 55  | inactive | Engineering | 130000  |
> > > | 10          | Judy  | 60  | active   | Sales       | 115000  |
> > >
> > > For a complex query like: “Find the top 3 departments by average salary among active employees over 40”
> > >
> > > POS decomposes the query into:
> > >
> > > ```markdown
> > > 1. Filter rows where age > 40
> > > 2. Filter rows where status = 'active'
> > > 3. Group by department
> > > 4. Compute AVG(salary)
> > > 5. Sort by that average
> > > 6. LIMIT 3
> > > ```
> > >
> > > Each step becomes one standalone SQL query whose result feeds the next step.
> > >
> > > However, we admit that **there’s one class of questions that SQLs can’t handle end-to-end**: free-form queries that require generating a narrative or summary. For example:
> > > “Summarize the basic information of the football clubs in Saint Petersburg.”
> > >
> > > For these, we use atomic SQL operations to transform the input table, then offload only the final writing step to an LLM to turn those results into free-form answers.
> > >
> > > We’ve added a discussion of atomic-step coverage: “Can every table-based question be decomposed into a set of atomic steps?” in `Appendix C` of our revised manuscript.

---

> > > > ### Author Response · Authors · 2025-05-06
> > > > **Authors' Response to Requested Change 2**
> > > >
> > > > > **Requested Change 2**: Why is SQL chosen instead of a general language like python?
> > > >
> > > > We would like to take the same example in **Requested Change 1** to demonstrate the reasons why we choose to use SQL rather than a general‐purpose language like Python for Table QA.
> > > >
> > > > Using Python, the program would be:
> > > > ```python3
> > > > # load the full table into memory
> > > > data = load_table("employees.csv")
> > > >
> > > > # Step 1: filter age > 40
> > > > step1 = [row for row in data if row["age"] > 40]
> > > >
> > > > # Step 2: filter status = "active"
> > > > step2 = [row for row in step1 if row["status"] == "active"]
> > > >
> > > > # Step 3: group by department and compute average salary
> > > > from collections import defaultdict
> > > > sums, counts = defaultdict(int), defaultdict(int)
> > > > for row in step2:
> > > >     dept = row["department"]
> > > >     sums[dept] += row["salary"]
> > > >     counts[dept] += 1
> > > > avg_salary = {dept: sums[dept] / counts[dept] for dept in sums}
> > > >
> > > > # Step 4: sort and take top 3
> > > > result = sorted(avg_salary.items(), key=lambda x: x[1], reverse=True)[:3]
> > > > print(result)
> > > > ```
> > > >
> > > > Using Python has the following drawbacks:
> > > > - We must pull the entire table into Python memory and loop over every row.
> > > > - Grouping and averaging require custom code; no built-in method (see Step 3).
> > > > - No built-in indexing—this can be very slow on large tables.
> > > >
> > > > Instead, SQL is **clear**, **highly optimized**, and **consistent**:
> > > > - **Clarity**: With SQL we can describe what we want (e.g. “rows where age > 40”), instead of using 3 steps sequentially: loop, filter > 40, and put to the list for Step 1. This makes queries more concise, self-documenting, and readable than writing Python code.
> > > > - **Speed & scalability**: Modern database engines have extensively optimized query planning, and indexing; they can process large tables far faster and more memory-efficiently than ad-hoc Python loops [3].
> > > > - **Consistency**: SQL’s syntax and semantics are stable across databases (e.g. Oracle, MySQL, Microsoft SQL Server, and PostgreSQL), so the same query runs everywhere. Python script, by contrast, require custom connectors, custom layers, and may vary with library versions
> > > >
> > > > [3] https://airbyte.com/blog/sql-vs-python-data-analysis, last access 05/03/2025.
> > > >
> > > >
> > > > We’ve added the discussion “SQL or Python; which is better for Table QA?” in `Appendix C` to explain why we chose SQL rather than Python.
> > > >
> > > > Please let us know if you have any further questions or concerns. We are more than happy to engage and clarify any points in our paper!

---

### Review · Reviewer_LDtj · 2025-05-14

**Summary Of Contributions:**

The paper introduces Plan-of-SQLs (POS), a new approach for making Table Question Answering (Table QA) more understandable. Instead of treating each question as a single, complex task, POS breaks it down into a series of simple, atomic steps which are converted into a straightforward SQL command, and run on the table one step at a time, making it easy to see how the final answer is reached. The key focus is on transparency. Unlike many other methods that rely on opaque, black-box models, every step in POS is clear and traceable, ensuring users can follow the reasoning behind each answer. Despite being simpler and more interpretable, POS still performs competitively on popular benchmarks like TabFact and WikiTQ.

**Audience:**

Yes

**Claims And Evidence:**

Yes

**Requested Changes:**

1) The paper will also benefit from a deeper discussion around the space of questions which cannot be broken down into sequential SQL queries and if we can auto-detect such cases.
2) Ablation study should also be carried out on performance on the framework so that we can pinpoint modules which are not necessary for neither interpretability nor performance.

**Strengths And Weaknesses:**

### Strengths

The paper is well written and it is easy to follow the arguments being made. Thorough evaluation using multiple datasets and interpretability measures, including both human and LLM-based assessments. The ablation study in Section 4.1.6 is very welcome and gives a lot of insight into breaking down the nature of the improvements. The efficiency analysis presented in the paper strengthens the novelty of the framework.

### Weaknesses
1) “POS uses a fallback that could diminish the program-only nature: if SQL execution fails, an LLM answers the question end-to-end—triggered in 3.16% of TabFact and 13.58% of WikiTQ samples, similar to other Table QA methods” - I think this significantly undermines the interpretability claim given in the paper. Although 13.58% already seems high, there is no guarantees on how high of a fraction will this be in real world datasets - which might prevent the use of this technique at scale.
2) The method is not able to handle tables where we cannot reasonably apply SQL like multi level nested tables etc. Although the authors do acknowledge this, most of the real world tables are of this format - and clearly end-to-end LLM reasoning will be strongly preferable over POS in those cases.

-----
Overall, although the applicability of the framework seems quite limited, I appreciate the strong analysis done by the authors and think that the TMLR community will find this interesting.

---

### Author Response · Authors · 2025-06-14
**Thank you, AE and the amazing anonymous reviewers.**

Dear Dr. Yuan,

Thank you very much for your support throughout the review process.

We’re grateful to you and the amazing anonymous reviewers for the constructive feedback and thoughtful handling of our submission. We’re delighted that the paper has been accepted and appreciate all the time and effort that went into the process.

Best regards,

Authors.

---

### Decision · Action_Editor_yiJg · 2025-06-11

**Recommendation:** Accept as is

**Additional Comments:**

All of the reviewer provides positive comments for the paper. The core contribution about interpretable LLM-based Table QA is generally an interesting problem for this community.

**Audience:**

Yes

**Audience Explanation:**

The paper discusses an interesting problem that would be interesting for the data management and broader machine learning community.

**Claims And Evidence:**

Yes

**Claims Explanation:**

The paper is technically solid, and the claims and statements in the paper are introduced with concrete evidence.